# Using CESM-RESFire to Understand Climate-Fire-Ecosystem Interactions and the Implications for Decadal Climate Variability

Yufei Zou[1†], Yuhang Wang[1], Yun Qian[2], Hanqin Tian[3], Jia Yang[4], Ernesto Alvarado[5]

[1]School of Earth and Atmospheric Sciences, Georgia Institute of Technology, Atlanta, GA 30332, USA.
[2]Atmospheric Sciences and Global Change Division, Pacific Northwest National Laboratory, Richland, WA 99354, USA.
[3]International Centre for Climate and Global Change Research, School of Forestry and Wildlife Sciences, Auburn University, AL 36849, USA.
[4]College of Forest Resources/Forest and Wildlife Research Center, Mississippi State University, MS 39762, USA.
[5]School of Environmental and Forest Sciences, University of Washington, Seattle, WA 98195, USA.
† Now at Pacific Northwest National Laboratory, Richland, WA 99354, USA.

*Correspondence to*: Yuhang Wang (yuhang.wang@eas.gatech.edu) and Yufei Zou (yufei.zou@pnnl.gov)

**Abstract.** Large wildfires exert strong disturbance to regional and global climate systems and ecosystems by perturbing radiative forcing as well as carbon and water balance between the atmosphere and land surface, while short- and long-term variations in fire weather, terrestrial ecosystems, and human activity modulate fire intensity and reshape fire regimes. The complex climate-fire-ecosystem interactions were not fully integrated in previous climate model studies, and the resulting effects on the projections of future climate change are not well understood. Here we use a fully interactive REgion-Specific ecosystem feedback Fire model (RESFire) that was developed in the Community Earth System Model (CESM) to investigate these interactions and their impacts on climate systems and fire activity. We designed two sets of decadal simulations using CESM-RESFire for present-day (2001-2010) and future (2051-2060) scenarios, respectively and conducted a series of sensitivity experiments to assess the effects of individual feedback pathways among climate, fire, and ecosystems. Our implementation of RESFire, which includes online land-atmosphere coupling of fire emissions and fire-induced land cover change (LCC), reproduces the observed Aerosol Optical Depth (AOD) from space-based Moderate Resolution Imaging Spectroradiometer (MODIS) satellite products and ground-based AErosol RObotic NETwork (AERONET) data and agrees well with carbon budget benchmarks from previous studies. We estimate the global averaged net radiative effect of both fire aerosols and fire-induced LCC at -0.59 $\pm$ 0.52 W m$^{-2}$, which is dominated by fire aerosol-cloud interactions (-0.82 $\pm$ 0.19 W m$^{-2}$), in the present-day scenario under climatological conditions of the 2000s. The fire-related net cooling effect increases by ~170% to -1.60 $\pm$ 0.27 W m$^{-2}$ in the 2050s under the conditions of the Representative Concentration Pathway 4.5 (RCP4.5) scenario. Such considerably enhanced radiative effect is attributed to the largely increased global burned area (+19%) and fire carbon emissions (+100%) from the 2000s to the 2050s driven by climate change. The net ecosystem exchange (NEE) of carbon between the land and atmosphere components in the simulations increases by 33% accordingly, implying that biomass burning is an increasing carbon source at short-term timescales in the future. High-latitude regions with prevalent peatlands would be more vulnerable to increased fire threats due to climate change and the increase of fire aerosols could counter the projected decrease of anthropogenic aerosols due to air pollution control policies in many

regions. We also evaluate two distinct feedback mechanisms that are associated with fire aerosols and fire-induced
LCC, respectively. On a global scale, the first mechanism imposes positive feedbacks to fire activity through enhanced
droughts with suppressed precipitation by fire aerosol-cloud interactions, while the second one manifests as negative
feedbacks due to reduced fuel loads by fire consumption and post-fire tree mortality and recovery processes. These
two feedback pathways with opposite effects compete at regional to global scales and increase the complexity of
climate-fire-ecosystem interactions and their climatic impacts.

## 1 Introduction

Large wildfires show profound impacts on human society and the environment with increasing trends in many regions
around the world during recent decades (Abatzoglou and Williams, 2016;Barbero et al., 2015;Clarke et al.,
2013;Dennison et al., 2014;Jolly et al., 2015;Westerling et al., 2006;Yang et al., 2011;Yang et al., 2015). They pose a
great threat to the safety of communities in the vicinity of fire-prone regions and distant downstream areas by both
destructive burning and increased health risks from fire smoke exposure. The global annual averaged premature deaths
due to fire smoke exposure was estimated about 339,000 (interquartile range: 260,000-600,000) during 1997 to 2006
(Johnston et al., 2012), while the total cost of fire-related socioeconomic burden would surge much higher if other
societal and environmental outcomes, such as morbidity of respiratory and cardiovascular diseases, expenditures of
defensive actions and disutility, and ecosystem service damages, were taken into account (Fann et al., 2018;Hall,
2014;Richardson et al., 2012;Thomas et al., 2017). In addition to hazardous impacts on human society, fire also exerts
strong disturbance to regional and global climate systems and ecosystems by perturbing radiation budget and carbon
balance between the atmosphere and land surface. In return, these short-term and long-term changes in fire weather,
terrestrial ecosystems, and human activity modulate fire intensity and reshape fire regimes in many climate change
sensitive regions. These processes were not fully included in previous climate model studies, increasing uncertainties
in the projections of future climate variability and fire activity (Flannigan et al., 2009;Hantson et al., 2016;Harris et
al., 2016;Liu et al., 2018). Most fire-related climate studies used a one-way perturbation approach by examining a
unidirectional forcing and response between climate change and fire activity without feedback. For instance, many
historical and future-projected fire responses to climate drivers were mainly based on offline statistical regression or
one-way coupled prognostic fire models in earth system models, while fire feedback to weather, climate, and
vegetation was neglected (e.g., Abatzoglou et al., 2019;Flannigan et al., 2013;Hurteau et al., 2014;Liu et al.,
2010;Moritz et al., 2012;Parks et al., 2016;Wotton et al., 2017;Young et al., 2017;Yue et al., 2013). The neglected
feedback could affect regional to global radiative forcing, biogeochemical and hydrological cycles, and ecological
functioning that may in turn modulate fire activity in local and remote regions (Harris et al., 2016;Liu, 2018;Pellegrini
et al., 2018;Seidl et al., 2017;Shuman et al., 2017). Similarly, climate studies (e.g., Jiang et al., 2016;Tosca et al.,
2013;Ward et al., 2012) that focused on climate responses to fire forcing used the same unidirectional approach but
from an opposite perspective, in which they evaluated multiple fire impacts on climate systems through fire aerosols,
greenhouse gases, and land albedo effects using climate sensitivity experiments with and without prescribed fire
emissions as model inputs. However, possible fire activity and emission changes in response to these fire weather and
climate variations were missing in such one-way perturbation modeling approaches.
To tackle these problems, we developed a two-way coupled RESFire model (Zou et al., 2019) with online land-
atmosphere coupling of fire-related mass and energy fluxes as well as fire-induced land cover change in CESM
(hereafter as CESM-RESFire). CESM-RESFire performs well using either offline observation-/reanalysis-based
atmosphere data or online simulated atmosphere, which is applied in this study to investigate the complex climate-
fire-ecosystem interactions as well as to project future climate change with fully interactive fire disturbance. In this
work, we use the state-of-the-science CESM-RESFire model to evaluate major climate-fire-ecosystem interactions
through biogeochemical, biogeophysical, and hydrological pathways and to assess future changes of decadal climate
variability and fire activity with consideration of these interactive feedback processes. We provide a brief model
description and sensitivity experiment settings in Section 2 and present modeling results and analyses on radiative
effects, carbon balance, and feedback evaluation in Section 3. Final conclusions and implications are followed in
Section 4.
**2 CESM-RESFire description, simulation setup, and benchmark data**
**2.1 Fire model and sensitivity simulation experiments**
RESFire (Zou et al., 2019) is a process-based fire model developed in the CESM version 1.2 modeling framework
that incorporates ecoregion-specific natural and anthropogenic constraints on fire occurrence, fire spread, and fire
impacts in both the CESM land component—the Community Land Model version 4.5 (CLM4.5) (Oleson et al., 2013)
and the atmosphere component—the Community Atmosphere Model version 5.3 (CAM5) (Neale et al., 2013). It is
compatible with either observation/reanalysis-based data atmosphere or the CAM5 atmosphere model with online
land-atmosphere coupling through aerosol-climate effects and fire-vegetation interactions. It includes two major fire
feedback pathways: the atmosphere-centric fire feedback through fire-related mass and energy fluxes and the
vegetation-centric fire feedback through fire-induced land cover change. These feedback pathways correspond to two
key climate variables, radiative forcing and carbon balance, through which fires exert their major climatic and
ecological impacts. Other features in CLM4.5 and CAM5, such as the photosynthesis scheme (Sun et al., 2012), the
3-mode modal aerosol module (MAM3; Liu et al., 2012), and the cloud microphysics (Morrison and Gettelman, 2008;
Gettelman et al., 2008) and macrophysics (Park et al., 2014) schemes, allow for more comprehensive assessments of
climate effects of fires through the interactions with vegetation and clouds. A simple treatment of secondary organic
aerosols (SOA) is used in CAM5 to derive SOA formation from anthropogenic and biogenic volatile organic
compounds (VOCs) with fixed mass fields (Table S1 in the Supplement). The total SOA mass is emitted as the SOA
(gas) species from the surface and then condensation/evaporation of gas-phase SOA to/from different aerosol modes
are calculated in the MAM3 module (Neale et al., 2013). The gas-phase photochemistry is not included in the CAM5
simulations, which precludes the possibility for evaluating chemistry-climate interactions. We also implement
distribution mapping-based online bias corrections for key fire weather variables (i.e., surface temperature,
precipitation, and relative humidity) to reduce negative influences of climate model biases in atmosphere simulation
and projection. Fire plume rise is globally universal parameterized based on atmospheric boundary layer height
(PBLH), fire radiative power (FRP), and Brunt-Väisälä frequency in the free troposphere (Sofiev et al., 2012). Please
refer to Zou et al. (2019) for more detailed fire model descriptions and to Sofiev et al. (2012) for the fire plume rise
parameterization. To quantify the impacts of fire-climate interactions under different climatic conditions, we designed
two groups of sensitivity simulations for present-day and future scenarios (Table 1). In each simulation group, we
conducted one control run (CTRLx, where x=1 or 2 indicates the present-day or future scenario, respectively) and two
sensitivity runs (SENSxA/B, where x is the same as that in CTRL runs and the notations of A and B are explained
below). The CTRL runs were designed with fully interactive fire disturbance such as fire emissions with plume rise
and fire-induced LCC with different boundary conditions for a present-day scenario (CTRL1; 2001-2010) and a
moderate future emission scenario (CTRL2) of the Representative Concentration Pathway 4.5 (RCP4.5; 2051-2060),
respectively. In each scenario, we turned off the atmosphere-centric feedback mechanisms (e.g., fire aerosol climate
effects) in SENSxA simulations (where x=1 or 2) and then turned off both atmospheric-centric and vegetation-centric
fire feedback (e.g., fire-induced LCC) in SENSxB simulations. Consequently, we estimated the atmosphere-centric
impacts of fire emissions on radiative forcing in the present-day scenario (RCP4.5 future scenario) by comparing
SENS1A (SENS2A) with CTRL1 (CTRL2). We also estimated the vegetation-centric impacts of fire-induced LCC
on terrestrial carbon balance in the present-day scenario (RCP4.5 future scenario) by comparing SENS1B (SENS2B)
with SENS1A (SENS2A). The net fire-related effects were evaluated by comparing CTRL runs with SENSxB runs
as both fire feedback mechanisms were turned off in the SENSxB runs. Using these sensitivity experiments, we are
able to evaluate two-way climate-fire-ecosystem interactions under the same integrated modeling framework that is
not possible in one-way perturbation studies considering either climate impacts on fires (Kloster et al., 2010;Kloster
et al., 2012;Thonicke et al., 2010) or fire feedback to climate (Jiang et al., 2016;Li et al., 2014;Ward et al., 2012;Yue
et al., 2015;Yue et al., 2016).
**2.2 Model input data**
We used the spun-up files from previous long-term runs (Zou et al., 2019) as initial conditions for the present-day
experiments (CTRL1 and SENS1A/B). The boundary conditions including the prescribed climatological (1981-2010
average) sea surface temperature and sea ice data for the present-day scenario were obtained from the Met Office
Hadley Centre (HadISST) (Rayner et al., 2003). Similarly, the nitrogen and aerosol deposition rates were also
prescribed from a time-invariant spatially varying annual mean file for 2000 and a time-varying (monthly cycle)
globally-gridded deposition file, respectively, as the standard datasets necessary for the present-day CAM5
simulations (Hurrell et al., 2013). The climatological 3-hourly cloud-to-ground lightning data via bilinear interpolation
from NASA LIS/OTD grid product v2.2 (http://ghrc.msfc.nasa.gov) 2-hourly lightning frequency data and the world
population density data were fixed at the 2000 levels for all the present-day simulations. The non-fire emissions from
anthropogenic sources (e.g., industrial, domestic and agriculture activity sectors) in the present-day scenario were
from the emission dataset (Lamarque et al., 2010) representing year 2000 for the Fifth Assessment Report of the
Intergovernmental Panel on Climate Change (IPCC AR5). Emissions of natural aerosols such as dust and sea salt were
calculated online (Neale et al., 2013), while vertically resolved volcanic sulfur and dimethyl sulfide (DMS) emissions
were prescribed from the AEROCOM emission dataset (Dentener et al., 2006).  Emission fluxes for the 5 VOC species
(isoprene, monoterpenes, toluene, big alkenes, and big alkanes) to derive SOA mass yields were prescribed from the
MOZART-2 dataset (Horowitz et al., 2003). For fire emissions, we replaced the prescribed GFED2 fire emissions
(van der Werf et al., 2006) from the default offline emission data with online coupled fire emissions generated by the
RESFire model in the CTRL runs. We then decoupled online simulated fire emissions in the SENS1A runs, in which
fire emissions were not transported to the CAM5 atmosphere model, to isolate the atmosphere-centric impacts of fire-
climate interactions. In both CTRL1 and SENS1A experiments, we allowed the semi-static historical LCC data for
the year 2000 from the version 1 of the Land-Use History A product (LUHa.v1) (Hurtt et al., 2006) to be affected by
post-fire vegetation changes (Zou et al., 2019). We then used the fixed LCC data for the year 2000 in the SENS1B
run and compared two SENS1 runs (SENS1A-SENS1B) to evaluate the vegetation-centric fire impacts on terrestrial
ecosystems and carbon balance in the 2000s.
For the future scenario experiments, we replaced all the present-day datasets with the RCP4.5 projection datasets
including the initial conditions and prescribed boundary conditions of global SST and sea ice data in 2050, the cyclical
non-fire emissions and deposition rates fixed in 2050 under the RCP4.5 scenario, and the annual LCC data for the
RCP4.5 transient period in 2050 based on the Future Land-Use Harmonization A products (LUHa.v1_future) (Hurtt
et al., 2006). All these datasets were described in the technical note of CAM5 (Neale et al., 2013) and stored on the
Cheyenne computing system (CISL, 2017) at the National Center for Atmospheric Research (NCAR)-Wyoming
Supercomputing Center (NWSC). It is worth noting that we used the present-day demographic data and observation-
based climatological lightning data in the future scenario given pathway-dependence and great uncertainties in future
projections of these inputs (Clark et al., 2017;Riahi et al., 2017;Tost et al., 2007;). In other words, we did not consider
the influence of fire ignition changes associated with human activity or lightning flash density in our future projection
simulations but focused on broad impacts of future climate change on fuel loads and combustibility as well as fire
weather conditions.
The global mean greenhouse gas (GHG) mixing ratios in the CAM5 atmosphere model were fixed at the 2000-year
levels ($CO_2$: 367.0 ppmv; $CH_4$:1760.0 ppbv; $N_2O$:316.0 ppbv) in all present-day experiments and they were replaced
by the prescribed RCP4.5 projection datasets with the well-mixed assumption and monthly variations in the future
scenarios. These GHG mixing ratios were then passed to the CLM4.5 land model in all sensitivity experiments. In
return, the land model provided the diagnostics of the balance of all carbon fluxes between net ecosystem production
(NEP, $g\ C\ m^{-2}\ s^{-1}$, positive for carbon sink) and depletion from fire emissions, landcover change fluxes, and carbon
loss from wood products pools, and then the computed net $CO_2$ flux was passed to the atmosphere model in forms of
net ecosystem exchange (NEE, $g\ C\ m^{-2}\ s^{-1}$). Though fire emissions could perturb the value of NEE at short-term scales,
it is often assumed that fire is neither a source nor a sink for $CO_2$ since fire carbon emissions are offset by carbon
absorption of vegetation regrowth over long-term scales (Bowman et al., 2009). Therefore, we did not consider the
radiative effect of fire-related GHGs in our sensitivity experiments. This kind of "concentration-driven" simulations
with prescribed atmospheric $CO_2$ concentrations for a given scenario have been used extensively in previous fire-
climate interaction assessments (e.g., Kloster et al., 2010;Li et al., 2014;Thonicke et al., 2010) and most of the RCP
simulations (Ciais et al., 2013).

## 2.3 Model evaluation benchmarks and datasets

Multiple observational and assimilated datasets were applied to evaluate the modeling performance regarding radiative forcing. We collected space-based column aerosol optical depth (AOD) from the level-3 MODIS Aqua monthly global product (MYD08_M3, Platnick et al., 2015) and ground-based version 3 aerosol optical thickness (AOT) level 2.0 data from the Aerosol Robotic Network (AERONET, https://aeronet.gsfc.nasa.gov/) project for comparison with the model simulated AOD data at 550 nm. The AERONET AOT at 550 nm were interpolated by estimating Ångström exponents based on the measurements taken at two closest wavelengths at 500 nm and 675 nm (see the Supplement for details). We then followed the Ghan method (Ghan, 2013) to estimate fire aerosol radiative effects ($RE_{aer}$) on the planetary energy balance in terms of aerosol-radiation interactions ($RE_{ari}$), aerosol-cloud interactions ($RE_{aci}$), and fire aerosol-related surface albedo change ($RE_{sac}$) in Eq. (1). The radiative effect related to fire-induced land cover change ($RE_{lcc}$) was estimated by comparing shortwave radiative fluxes at the top-of-atmosphere (TOA) between SENSxA (with fire-induced LCC) and SENSxB (without fire-induced LCC) experiments. By summing up all these terms, we estimated the fire-related net radiative effect ($RE_{fire}$) as the shortwave radiative flux difference between CTRLx (with fire aerosols and fire-induced LCC) and SENSxB (without fire aerosols and fire-induced LCC) experiments:

$$RE \text{ of interaction of radiation with fire aerosol}: RE_{ari} = \Delta(F - F_{clean})$$
$$RE \text{ of interaction of clouds with fire aerosol}: RE_{aci} = \Delta(F_{clean} - F_{clear,clean})$$
$$RE \text{ of surface albedo change induced by fire aerosol}: RE_{sac} = \Delta F_{clear,clean}$$
$$net\ RE \text{ of fire aerosol}: RE_{aer} = RE_{ari} + RE_{aci} + RE_{sac} = F_{CTRLx} - F_{SENSxA}$$
$$RE \text{ of fire induced land cover change}: RE_{lcc} = F_{SENSxA} - F_{SENSxB}$$
$$net\ RE \text{ of fire}: RE_{fire} = RE_{aer} + RE_{lcc} = F_{CTRLx} - F_{SENSxB}$$

$$(1)$$

where $\Delta$ is the difference between control and sensitivity simulations, $F$ is the shortwave radiative flux at the TOA, $F_{clean}$ is the radiative flux calculated as an additional diagnostics from the same simulations but neglecting the scattering and absorption of solar radiation by all aerosols, and $F_{clear,clean}$ is the flux calculated as additional diagnostic but neglecting scattering and absorption by both clouds and aerosols. The surface albedo effect is largely the contribution of changes in surface albedo induced by fire aerosol deposition and land cover change, which is small but non-negligible in some regions (Ghan, 2013). We used similar modeling settings including the 3-mode modal aerosol scheme (MAM3) (Liu et al., 2012) and the Snow, Ice, and Aerosol Radiative (SNICAR) module (Flanner and Zender, 2005) and compared our online coupled fire modeling results against previous offline prescribed fire modeling studies (Jiang et al., 2016;Ward et al., 2012) in the next section.

We also examined the modeling performance on burned area and terrestrial carbon balance such as fire carbon emissions, gross primary production (GPP, g C m$^{-2}$ s$^{-1}$, positive for vegetation carbon uptake), net primary production (NPP, g C m$^{-2}$ s$^{-1}$, positive for vegetation carbon uptake), net ecosystem productivity (NEP, g C m$^{-2}$ s$^{-1}$, positive for net ecosystem carbon uptake), and net ecosystem exchange (NEE, g C m$^{-2}$ s$^{-1}$, positive for net ecosystem carbon emission). The model simulated burned area and fire carbon emissions were evaluated against the satellite based GFED4.1s datasets (Giglio et al., 2013;Randerson et al., 2012;van der Werf et al., 2017), and these carbon budget related variables were calculated in Eqs. (2) and (3) and compared with the MODIS primary production products (Zhao et al., 2005;Zhao and Running, 2010), previous modeling results used for terrestrial model comparison projects

(Piao et al., 2013) and the IPCC AR5 report (Ciais et al., 2013), and the global carbon budget assessment (Le Quere
et al., 2013) by the broad carbon cycle science community.
$\quad$ GPP = NPP + $R_a$ = (NEP + $R_h$) + $R_a$, $\hfill$ (2)
$\quad$ NEE = $C_{fe}$ + $C_{lh}$ − NEP = $C_{fe}$ + $C_{lh}$ + $R_h$ + $R_a$ − GPP, $\hfill$ (3)
where $R_a$ is the total ecosystem autotrophic respiration (g C m$^{-2}$ s$^{-1}$), $R_h$ is the total heterotrophic respiration (g C m$^{-2}$
s$^{-1}$), $C_{fe}$ is the fire carbon emissions (g C m$^{-2}$ s$^{-1}$), and $C_{lh}$ is the carbon loss (g C m$^{-2}$ s$^{-1}$) due to land cover change,
wood products, and harvest.

## 3 Modeling results and discussion

### 3.1 Evaluation of fire-related radiative effects

Figure 1 shows the comparison of the model simulated 10-year annual averaged column AOD at 550nm from CTRL1
and space-based AOD from MODIS aboard the Aqua satellite. It's noted that both AOD data result from all sources
including fire and non-fire emissions, and significant differences exist in specific regions due to large biases in model
emission inputs and aerosol parameterization. In the MODIS AOD data, the most noticeable hotspot regions include
eastern China, South Asia such as India, and Africa. The first two regions are contributed mostly by anthropogenic
emissions, while the last one is dominated by fire emissions. Since the non-fire emissions used in CAM5 simulations
are 2000-based (Lamarque et al., 2010) and low biased comparing to rapid emission increases in many Asian
developing countries (Kurokawa et al., 2013), the simulated hotspot regions in East and South Asia are not as
appreciable as those observed in the remote sensing data. The model results also show underestimation in rainforests
over South America and Central Africa, where large fractions of aerosols are contributed by primary and secondary
organic aerosols from biogenic sources and precursors (Gilardoni et al., 2011) that are missing in the simulation.
Another possible cause for the underestimation problem is underrepresented burning activity due to deforestation and
forest degradation and consequently underestimated fire aerosols emissions in these regions. The AOD simulations
over tropical savanna regions with pervasive biomass burning activities are also lower than the satellite observations,
which might be attributable to both underestimated online fire emissions and too strong wet scavenging of primary
carbonaceous aerosols in the CAM5-MAM3 model (Liu et al., 2012). The CAM5 model overestimates dust emissions
significantly with some spuriously high AOD hotspots emerging over the Sahara, Arabian, South Africa, and Central
Australia desert regions. This dust AOD overestimation problem was also found in a previous dust modeling study
using the release version of the CAM5-MAM3 model (Albani et al., 2014).
$\quad$ To further evaluate the fire-related AOD modeling performance, we compare the difference between CTRL1 and
SENS1A to isolate aerosol contributions from fire sources in Fig. 2. The spatial distribution of fire-related AOD
clearly highlighted African savanna as a major biomass burning region. We also compare monthly AOD at six fire-
prone regions with AERONET observations to get a better understanding of temporal variations of fire aerosols. Most
sites show strong seasonal variations in monthly AOD as observed by AERONET, and the CESM-RESFire model
well capture fire seasonality in these regions. Generally, the model AOD results are at the lower ends of the uncertainty
ranges of ground-based observations in most regions due to limited spatial representativeness of coarse model grid
resolution and fire emissions, especially over African savannas like Ilorin (Fig. 2e) and Southeast Asian rainforests
like Jambi (Fig. 2g) where agricultural and deforestation related burning activity prevails.
Lastly, we estimate present-day radiative effects of fire aerosols and fire-induced land cover change and compare
the results with previous studies in Fig. 3 and Table 2. The radiative effect of fire aerosol-radiation interactions ($RE_{ari}$)
is most prominent in tropical Africa and downwind Atlantic Ocean areas as well as South America and eastern Pacific.
High-latitude regions like eastern Siberia also show significant positive radiative effects due to fire emitted light
absorbing aerosols such as black carbon (BC). The land-sea contrast of radiative warming and cooling effects over
Africa and South America are attributed to differences of cloud cover fractions over land and ocean areas (Jiang et al.,
2016). In these regions, cloud fractions and liquid water path are much larger over downwind ocean areas than land
areas during the fire season. Cloud reflection of solar radiation strongly enhances light absorption by fire aerosols
residing above low-level marine clouds (Abel et al., 2005; Zhang et al., 2016).
The radiative effect of fire aerosol-cloud interactions ($RE_{aci}$) shows generally cooling effects in most regions due to
scattering and reflections by enhanced cloudiness, and these cooling effects are more pervasive over high-latitude
regions such as boreal forests in North America and eastern Siberia. The land-sea contrast of radiative effects emerges
again in the vicinity of Africa and South America, but the signs of the contrasting effect related with aerosol-cloud
interactions are opposite to these from aerosol-radiation interactions. The large amounts of fire aerosols suppress low-
level clouds over the African land region by stabilizing the lower atmosphere through reduction of radiative heating
of the surface. However, fire aerosols increase cloud cover and brightness in the downwind Atlantic Ocean areas
because they increase the number of cloud condensation nuclei and the larger cloud droplet number density reduce
cloud droplet sizes (Lu et al., 2018; Rosenfeld et al., 2019; Fig. S1 in the Supplement). The radiative effect of fire
aerosol-related surface albedo change ($RE_{sac}$) shows contrasting radiation effects with strong warming effects over
most Arctic regions caused by deposition of light-absorbing aerosol over ice and snow and reduction of surface albedo,
but moderate cooling effects in boreal land regions such as Canada and eastern Siberia, which are related to fire
aerosol-induced snowfall and snow cover change and associated surface albedo change (Ghan, 2013; Fig. S2 in the
Supplement). Besides spatial heterogeneity in fire-induced radiative effects, these radiative effects also show
significant temporal variations that are related with fire seasonality. Figure 4 shows zonal averaged time-latitude cross
sections of fire aerosol emissions and fire-induced changes in clouds and radiative effects. Massive fire carbonaceous
emissions shift from the Northern Hemisphere tropical regions in boreal winter to the Southern Hemisphere tropical
regions in boreal summer, when similar amounts of fire emissions are also observed in boreal mid- and high-latitude
regions (Fig. 4a/b). Fire aerosols greatly increase cloud condensation nuclei (CCN, Fig. 4c) and cloud droplet number
concentrations (CDNUMC, Fig. 4d) in these regions, while the increase in cloud water path (CWP, Fig. 4e) and low
cloud fraction (CLDLOW, Fig. 4f) are more significant in boreal high-latitude regions than in the tropics.  The low
solar zenith angle in high-latitude regions enhances solar radiation absorption by light-absorbing aerosols and results
in stronger changes in radiative effects by aerosol-radiation interactions during boreal summer (Fig. 4g). In the
meantime, increased CWP and CLDLOW in high-latitude regions also lead to much stronger cooling effects by
aerosol-cloud interactions ($RE_{aci}$) (Fig. 4h), which overwhelm the increase in $RE_{ari}$. These modeling results based on
the online coupled RESFire model show similar spatiotemporal patterns with these in Jiang et al. (2016), which used
the same version of the CAM5 atmosphere model with a 4-mode modal aerosol module (MAM4) that was driven by
offline prescribed fire emissions.

In general, the 10-year averaged global mean values and standard deviations of interannual variations for fire

aerosol-related $RE_{ari}$, $RE_{aci}$, and $RE_{sac}$ in the 2000s are -0.003 $\pm$ 0.013 W m$^{-2}$, -0.82 $\pm$ 0.19 W m$^{-2}$, and 0.19 $\pm$ 0.61 W
m$^{-2}$, respectively, and fire-induced $RE_{lcc}$ is 0.04 $\pm$ 0.38 W m$^{-2}$. After combining all these forcing terms, we estimate a
net $RE_{fire}$ of -0.59 $\pm$ 0.51 W m$^{-2}$ for the present-day scenario that is larger than the estimate of -0.55 W m$^{-2}$ in the
previous fire radiative effect studies (Jiang et al., 2016;Ward et al., 2012). It is noted that both Ward et al. (2012) and
Jiang et al. (2016) used prescribed fire emissions from CLM3 model simulations (Kloster et al., 2010;Kloster et al.,
2012) and GFED datasets (Giglio et al., 2013;Randerson et al., 2012), respectively, for their uncoupled fire sensitivity
simulations. The annual fire carbon emissions used by Ward et al. (2012) ranged from 1.3 Pg C yr$^{-1}$ for the present-
day simulation to 2.4 Pg C yr$^{-1}$ for the future projection with ECHAM atmospheric forcing, while the fire BC, POM
and SO$_2$ emissions used by Jiang et al. (2016) were based on the GFEDv3.1 dataset with an annual averaged fire
carbon emission of 1.98 Pg C yr$^{-1}$ (Randerson et al., 2012). Their fire emissions are lower than the RESFire model
simulation of 2.6 Pg C yr$^{-1}$ (Table 3) in this study, which contribute to the differences in the estimates of fire aerosol
radiative effects. It is also worth noting that all fire emissions were released into the lowest CAM level as surface
sources by Ward et al. (2012), and a default vertical profile of fire emissions based on the AEROCOM protocol
(Dentener et al., 2006) was used by Jiang et al. (2016) in their CAM5 simulations. In our simulations, we used a
simplified plume rise parameterization (Sofiev et al., 2012) based on online calculated fire burning intensity (FRP)
and atmospheric stability conditions (PBLH and Brunt-Väisälä frequency) in CESM-RESFire and applied vertical
profiles with diurnal cycles to the vertical distribution of fire emissions. The simulations of annual median heights of
fire plumes for the present-day and RCP4.5 future scenarios are shown in Fig. 5. Previous observation-based injection
height studies suggested that only 4–12% fire plumes could penetrate planetary boundary layers with most fire plumes
stay within the near surface atmosphere layers (val Martin et al., 2010). Our plume-rise simulation results agree with
these estimates, though a quantitative comparison is beyond the scope of this study because of the inconsistency
between simulated and actual meteorological conditions. It is also noted that there is no systematic change in plume
rise height distributions between the RCP4.5 future scenario and present-day scenarios, both of which show most fire
plumes (~80%) rise less than 1000 m. Comparing to surface released fire emissions in previous studies (Ward et al.,
2012), our higher elevated fire plumes affect the vertical distribution and lifetime of fire aerosols and further influence
regional radiative effects after long-range transport of fire aerosols.
**3.2 Fire-related disturbance to carbon balance**
In addition to the atmosphere-centric fire-induced radiative effects, we also quantify the vegetation-centric terrestrial
carbon budget changes to evaluate fire disturbance to terrestrial ecosystems. We use the previous model inter-
comparison studies and the latest GFEDv4.1s datasets as evaluation benchmarks and examine fire-related metrics
including global burned area and fire carbon emissions (Fig. 6 and Table 3). We also collect global scale GPP, NPP,
and NEE from previous literatures (Ciais et al., 2013; Piao et al., 2013;Zhao and Running, 2010) to compare with our
simulation results (Table 3). The RESFire model performs well in global burned area and fire carbon emissions driven
by either offline observation-/reanalysis-based CRUNCEP atmosphere data (RESFire_CRUNCEP) and online CAM5
simulated atmosphere data after bias corrections (RESFire_CAM5c). The annual averaged burned area results of both
RESFire_CRUNCEP (508 $\pm$ 15 Mha yr$^{-1}$) and RESFire_CAM5c (472 $\pm$ 14 Mha yr$^{-1}$) are very close to the GFEDv4.1s
benchmark value of 510 $\pm$ 27 Mha yr$^{-1}$, while the default fire model in CLM (322 Mha yr$^{-1}$) is significantly low biased.
For fire carbon emissions, the offline RESFire_CRUNCEP result (2.3 $\pm$ 0.2 Pg C yr$^{-1}$) agrees well with the
GFEDv4.1s benchmark of around 2.2 $\pm$ 0.4 Pg C yr$^{-1}$, and the online RESFire_CAM5c result shows a 18% higher
value (2.6 $\pm$ 0.1 Pg C yr$^{-1}$) than the benchmark. Since the GFED emission datasets are low biased due to low satellite
detection rates for small fires under canopy and clouds, previous fire studies (Johnston et al., 2012;Ward et al., 2012)
rescaled fire emissions in their practice for climate and health impact assessment. Here, a moderate increase in online
estimated fire carbon emissions would reduce the need for fire emission rescaling. Such difference is also consistent
with the changes in different versions of the GFED datasets, which show a 11% increase of global fire carbon
emissions in the latest GFED4s as compared with the old GFED3 for the overlapping 1997-2011 time period (van der
Werf et al., 2017). This increased global fire carbon emissions in the GFED4s dataset result from a substantial increase
in global burned area (+37%) due to inclusion of small fires and a modest decrease in mean fuel consumption (-19%)
according to van der Werf et al. (2017). Since carbon emissions from deforestation fires and other land use change
processes are a key component to estimate global carbon budget (Le Quere et al., 2013), improved fire emission
estimation would benefit carbon budget simulation in the land model.
We then compare the CLM simulated carbon budget variables such as GPP and NEE against 10 process-based
terrestrial biosphere models that were used for the IPCC fifth Assessment Report (Piao et al., 2013). Both the offline
and online CLM GPP results are around 142 Pg C yr$^{-1}$, which are higher than the MODIS primary production products
(MOD17) of 109.29 Pg C yr$^{-1}$ (Zhao et al., 2005) and near the upper bound of ensemble modeling results (133 $\pm$ 15
Pg C yr$^{-1}$) (Piao et al., 2013). Such high GPP estimation leads to ~11% higher NPP in the CLM simulations than the
MODIS global average annual NPP product of 53.5 Pg C yr$^{-1}$ from 2001 to 2009 (Zhao and Running, 2010) as well
as the old modeling result (54 Pg C yr$^{-1}$) based on the default fire model in CLM developed by Li et al. (2013;2014)
(hereafter as CLM-LL2013). These differences may result from the different atmosphere forcing data used to drive
the CLM land model. However, the NEE results based on the CESM-RESFire model are consistent with the
benchmarks from the IPCC AR5 (Ciais et al., 2013) and ensemble modeling results (Piao et al., 2013), indicating a
good land modeling performance with online fire disturbance in CESM.
After the evaluation of carbon budget in the CLM land model, we further decompose the components in NEE and
compare the new CESM-RESFire simulation results with previous fire model simulations by Li et al. (2014).
Following their experiment setting in Li et al. (2014), we isolate fire contributions to each carbon budget variables by
differencing the fire-on and fire-off experiments driven by the CRUNCEP data atmosphere in Table 4. We find a 58%
increase in fire-induced NEE variations simulated by CESM-RESFire than CLM-LL2013. This increase is attributed
to enhanced fire emissions and suppressed NEP in CESM-RESFire. As discussed in the previous section, CESM-
RESFire simulates higher annual averaged fire carbon emissions (2.08 Pg C yr$^{-1}$) than CLM-LL2013 (1.9 Pg C yr$^{-1}$),
which contributes 31% of the difference in their NEE changes. Furthermore, CESM-RESFire simulates smaller NEP
changes due to fire disturbance, which is attributable to fire-induced land cover change in RESFire. Fire-induced

whole plant mortality and post-fire vegetation recovery are implemented in the new CESM-RESFire model (Zou et al., 2019), both of which are not included in the default CLM-LL2013 model. The newly incorporated fire-induced land cover change would influence ecosystem productivity and respiration as shown by carbon budget variables in Table 4. Specifically, the fire-induced whole plant mortality and recovery would moderate the variations in ecosystem productivity and respiration and further suppress fire-induced NEP changes. The suppressed NEP change explains 52% of the total difference between CESM-RESFire and CLM-LL2013 in simulated NEE changes.

Similar suppression effects of fires on NEP were also found in Seo and Kim (2019), in which they used the CLM-LL2013 fire model but enabled the dynamic vegetation (DV) mode to simulate post-fire vegetation changes. Though the DV mode of the CLM model is capable of simulating vegetation dynamics, considerable biases exist in the online simulation of land cover change by the coupled CLM-DV model (Quillet et al., 2010) and may undermine the interpretation of fire-related ecological effects. For instance, the global fractions of bare ground and needleleaf trees in the CLM-DV simulations are much larger than these in the non-DV (BGC only) simulation in Seo and Kim (2019), while the fractions of shrub and broadleaf trees with active DV are less than these without DV regardless of whether fire disturbance are included or not in the simulations. These biases could distort ecosystem properties such as primary production and carbon exchange as well as fire-related ecological effects.

Similar to fire-related radiative effects, we examine changes of carbon budget variables in the RCP4.5 future scenario in Table 5 and Fig. 7. The global burned area increases by 19% from the present-day scenario in CTRL1 (464 $\pm$ 19 Mha yr$^{-1}$) to the RCP4.5 future scenario in CTRL2 (551 $\pm$ 16 Mha yr$^{-1}$) (Fig. 7a). Accordingly, the annual averaged fire carbon emission increases by 100% from 2.5 $\pm$ 0.1 Pg C yr$^{-1}$ at present to 5.0 $\pm$ 0.3 Pg C yr$^{-1}$ in the future (Fig. 7b). This increase is larger than a previous CLM simulated result of 25%~52% by Kloster et al. (2010;2012), which might result from different climate sensitivity between CESM-RESFire and the old fire model in CLM. It's noted that recent satellite-based studies found decreasing trends in burned area over specific regions such as Northern Hemisphere Africa driven by human activity and agricultural expansion (Andela and van der Werf, 2014; Andela et al., 2017). Though we mainly focus on fire-climate interactions without consideration of human impacts in this study, the RESFire model is capable of capturing the anthropogenic interference on fire activity and reproducing observation-based long-term trends of regional burning activity driven by climate change and human factors (Zou et al., 2019). The carbon budget variables including GPP, NEP, and NEE increase by 4%, 7%, and 33%, respectively (Fig. 7c-d). These carbon variables affect terrestrial ecosystem productivity as well as fuel load supply for biomass burning, which further modulate fire emissions that lead to discrepancies between burned area and emission changes. For instance, most decreasing changes in burned area occur in tropical and subtropical savannas and grasslands, while significant increasing changes are evident in boreal forest and tropical rainforests of Southeast Asia (Fig. 7a). This spatial shift of burning activity from low fuel loading areas (e.g., grassland) to high fuel loading areas (e.g., forest) greatly amplifies the changes in fire emissions due to boosted fuel consumption. The complex climate-fire-ecosystem interactions will be discussed in the next section.

**3.3 Simulations of climate-fire-ecosystem interactions using CESM-RESFire**

In the last section, we find a 19% increase of global burned area in the RCP4.5 future scenario comparing with the present-day scenario. We then examine spatial distributions and driving factors of this change in Fig. 8. The fire ignition distribution shows heterogeneous changes with significant increases in boreal forest regions over Eurasia as well as rainforest regions in South America but decreases in South American savanna and African rainforests and savanna. These changes in fire ignition are mainly driven by changes in fuel combustibility as shown by fire combustion factors (Fig. 8b), which are computed using fire weather conditions including 10-day running means of surface air temperature, precipitation, and soil moisture (Zou et al., 2019). The spatial distribution changes of fire spread (Fig. 8c) shows similar but more apparent patterns of increased fire spread rates over most regions except savanna and rainforests in Africa and South America, which are attributed to the changes in fire spread factors (Fig. 8d). These fire spread factors depend on surface temperature, relative humidity, soil wetness, and wet canopy fractions that modulate fuel moisture and fire spread rates in the model (Zou et al., 2019). The burned area changes are driven by changes of fire weather conditions affecting both fire ignition and fire spread, with a global spatial correlation coefficient of 0.4 between differences in fractional burned area (Fig. 7a) and fire counts (Fig. 8a) and of 0.38 between burned area (Fig. 7a) and fire spread rates (Fig. 8c). These burning activity changes found in this study also agree quite well with previous long-term projections based on an empirical statistical framework and a multi-model ensemble of 16 GCMs, in which they found good model agreement on increasing fire probabilities (~62%) at mid- to high-latitudes as well as decreasing fire probabilities (~20%) in the tropics (Moritz et al., 2012).

To understand the changes in specific fire weather variables, we compare the differences of surface air temperature, total precipitation rates, relative humidity, and surface wind speed between the future (CTRL2) and present-day (CTRL1) scenarios in Fig. 9. As expected in a modest warming scenario, the global annual mean temperature is projected to increase by 1.7 °C on average with pervasive warming over land areas (Fig. 9a). The temperature increases are stronger in high latitude regions like Alaska, northern Canada, and Antarctica as well as Australia. Meanwhile, hydrological conditions also undergo significant but nonhomogeneous changes in many regions in the projection, with hot and dry weather conditions favorable for fire in Australia, Southeast Asia, Central America, and the northern coast of South America (Fig. 9b and 9c). Most of these regions also show increased surface wind speed that is conducive to faster fire spread (Fig. 9d). Since these variations in fully coupled CTRL experiments can be induced by either global warming driven weather changes or fire feedback, we further decompose the total changes into two components: one without fire feedback (i.e., SENS2B-SENS1B) and the other purely by fire feedback (i.e., (CTRL2-CTRL1)-(SENS2B-SENS1B)). We show the fire induced weather changes in Fig. 10 and these without fire feedbacks in Fig. S3 in the Supplement. It is clear that the majority of the changes in fire weather conditions is driven by atmospheric conditions associated with global warming since the spatial patterns in Fig. 9 and Fig.S3 almost resemble each other over most land regions. However, fire feedbacks also exert nonnegligible effects to local and remote weather conditions that manifest as positive or negative feedback mechanisms to regional fire activities. For instance, Australia shows increased temperature (Fig. 10a) and surface wind speed (Fig. 10d), and decreased precipitation (Fig. 10b) and relative humidity (Fig. 10c) induced by fire, which are consistent with these changes without fire feedbacks (Fig. S3 in the Supplement) or the total changes (Fig. 9). In contrast, most Eurasian regions show decreased temperature (Fig.

10a) and increased relative humidity (Fig. 10c), with nonhomogeneous changes of precipitation (Fig. 10b) in response
to fire perturbations. These regionally varying results suggest complex interactions between fire and climate systems
that merit further investigation.
Therefore, we aggregate regional burned areas in each experiment and compare their changes between the two
scenarios to quantify regional effects of different feedback mechanisms (Fig. 11). An atmosphere-centric feedback
pathway is identified by comparing relative changes of regional burned area with (i.e., CTRL2-CTRL1) and without
(i.e., SENS2A-SENS1A) fire aerosol effects, while a vegetation-centric feedback pathway is identified by comparing
relative changes of regional burned area with (i.e., SENS2A-SENS1A) and without (i.e., SENS2B-SENS1B) fire
induced LCC. The comparison of relative changes in regional burned area with different feedback pathways reveal
distinct regional responses to these fire related atmospheric and vegetation processes. The most significant fire
feedback effects occur in North America (Fig. 11a) and South America (Fig. 11b), with the former dominated by
negative vegetation-centric fire feedback and the latter dominated by positive atmosphere-centric fire feedback. By
including fire induced LCC, the projected burned area increases over North America in the 2050s are greatly
suppressed and reduced from +172% in SENS2B to +94% in SENS2A and +93% in CTRL2, respectively. In contrast,
the burned area increases over South America considerably enlarges after incorporating fire aerosol effects in the
projection, from +112% in SENS2A and +113% in SENS2B to +142% in CTRL2. The fire feedback effects are also
evident in many other regions, such as similar positive atmosphere-centric feedbacks in Southeast Asia (Fig. 11g) and
Oceania (Fig. 11h) but negative atmosphere-centric feedbacks in Africa (Fig. 11e and 11f). The signs of these feedback
effects are determined by fire perturbation on regional fuel and fire weather conditions such as precipitation through
fire aerosol-cloud-precipitation interactions or changed vegetation evapotranspiration due to fire induced LCC (Fig.
S5 in the Supplement). It's worth noting that these feedback effects could enhance (e.g., North America and Southeast
Asia) or compensate (e.g., Northern Hemisphere and Southern Hemisphere Africa) each other in different regions,
which further increase the complexity of climate-fire-ecosystem interactions at regional and global scales. On a global
average, the net effect of fire feedbacks is almost neutral (Fig. 11i and Table 5) due to the offsetting between positive
vegetation-centric and negative atmosphere-centric feedbacks, which are largely dominated by burning activity in
African regions.
Lastly, we compare the difference of climate radiative forcing associated with these burning activity changes between
the future and present-day scenarios in Table 2 and Fig. 12. Due to broadly increased burning activities in the future
projection, fire aerosols are strongly enhanced over most fire-prone regions except Northern Hemisphere Africa and
South Asia (Fig. 12a), where the projected burning activity is suppressed as discussed in previous sections. Increased
fire aerosols lead to diverse responses in cloud liquid water path, with large increases in high-latitude regions but
generally decreases in the tropics and sub-tropics (Fig. 12b). These fire and weather changes result in pronounced
responses in radiative forcing through multiple pathways including aerosol-radiation interaction (Fig. 12c), aerosol-
cloud interaction (Fig. 12d), and fire induced LCC (Fig. 12e). The fire aerosol related RE changes show more
consistent and statistically significant changes over fire-prone regions than these induced by LCC. Previous studies
have suggested a net cooling effect of deforestation that could compensate for GHG waring effects on a global scale
(Bala et al., 2007;Jin et al., 2012;Randerson et al., 2006). Though our model captures the reduction of forest coverage
and increased springtime albedo in high-latitude regions (Fig. S6 in the Supplement), the radiative effect of fire
induced LCC is almost neutral on a global basis in both present-day and future scenarios (Table 2). In general, most
burning regions with increased fire aerosols show cooling effects due to enhanced aerosol scattering of solar radiation,
while those with decreased fire aerosols show warming effects (Fig. 12c). Fire aerosol direct radiative forcing is
overwhelmed by much stronger indirect effects through aerosol-cloud interactions (Fig. 12d), with pervasive cooling
effects in high-latitude regions with increased cloudiness (Fig. 12b). Such indirect effects also dominate the net fire
radiative effects at both regional and global scales, contributing to a 171% increase of global net fire radiative effect
in the RCP4.5 future scenario (Table 2). This projection result is larger than the change in net fire radiative forcing
based on the CCSM future projection in Ward et al. (2012), which suggested a 51% increase from -0.55 W m$^{-2}$ in the
2000s to -0.83 W m$^{-2}$ in the 2100s (Table 2). It is noted that their net estimate of fire radiative forcing changes includes
other offline-based fire climate effects such as fire-related GHGs impacts and climate-biogeochemical cycle
feedbacks, which could dampen the cooling effect of fire aerosols.
**3.4 Discussion of modeling uncertainties**
As discussed in previous sections, the complex climate-fire-ecosystem interactions in fire related atmospheric and
vegetation processes can introduce large uncertainties in the fire projections and associated climate effects. Here we
list major uncertainty sources that deserve further investigations in the future.
(1) Future projection of fire triggers such as lightning and human activity is highly uncertain and difficult to
explicitly parameterize in global climate models at present. Previous studies suggested different and even
contradictory changes in projected lightning in the future (Clark et al., 2017; Finney et al., 2017) due likely
to the difference in lightning parameterization schemes used. Pathway dependent long-term projections of
demographic data and socioeconomic conditions are also highly uncertain (Riahi et al., 2017). For these
reasons, we did not consider these factors in our projection experiments by using fixed demographic and
lightning data. Assessing the impacts of these factors will require implementations of different lightning
parameterizations and socioeconomic scenarios in climate simulations.
(2) Similar uncertainties arise from future projections of land use and land cover changes and dynamic global
vegetation modeling (DGVM). These anthropogenic and ecological processes could directly or indirectly
modulate fire activities by changing fire risks and fuel availability. In this study, we used semi-static land use
and land cover data with the sole consideration of fire perturbations in both historical and projection
scenarios. The inclusion of DGVM will enable the projection of vegetation distributions but introduce
additional uncertainties (Zou et al., 2019).
(3) The uncertainties of fire emission estimates arise from those in surface fuel loads, combustion completeness,
emission factors, and vertical distributions with rising fire plumes. More measurements of these parameters
over extended temporospatial scales are needed to fully evaluate these terms in the fire models. A newly
developed fire plume rise scheme (Ke et al., 2019) has been recently implemented in the fire model used in
this study and will be used for future fire modeling and evaluation studies.
(4)  Last but not the least, fire aerosol radiative effects and aerosol-cloud interactions play an important role in

simulating the climate effects of fire aerosols. Though the atmosphere model used in this study incorporates

aerosol-cloud interactions, these atmospheric processes across multiple spatial and temporal scales are major

contributors to the uncertainties of the climate change assessments (Ciais et al., 2013; Seinfeld et al., 2016).

Community wide efforts are ongoing to quantify and reduce the uncertainties of climate modeling discussed

above.

**4 Conclusions and implications**
In this study, we conducted a series of fire-climate modeling experiments for the present-day and future scenarios with
explicit implementation of multiple climate-fire-ecosystem feedback mechanisms. We evaluated the CESM-RESFire
modeling performance in the context of fire-related radiative effects and terrestrial carbon balance. Various fire
radiative effects for the present-day and the RCP4.5 future scenarios are summarized in Fig. 13. We focus on radiative
forcing changes related with fire aerosols and fire-induced land cover change. We find an enhanced net fire radiative
effect, which is caused by increased global burning activity and subsequent aerosol-cloud interactions, increasing from
$-0.59 \pm 0.51$ W m$^{-2}$ in the 2000s to $-1.60 \pm 0.27$ W m$^{-2}$ in the 2050s. Annual global burned area and fire carbon
emissions increase by 19% and 100%, respectively, with large amplifications in boreal regions due to suppressed
precipitation and enhanced fire ignition and spread rates. These changes imply increasing fire danger over high-
latitude regions with prevalent peat lands, which will be more vulnerable to increased fire threats due to climate
change. Potential increasing burning activity in these regions may greatly increase fire carbon and tracer gas and
aerosol emissions that could have enormous impacts on terrestrial carbon balance and radiative budget. Our modeling
results imply that the increase of fire aerosols could compensate the projected decrease of anthropogenic aerosols due
to air pollution control policies in many regions (e.g., the eastern U.S. and China) (EPA, 2019;McClure and Jaffe,
2018;Wang et al., 2017;Zhao et al., 2014), where significant aerosol cooling effects dampen GHG warming effects
(Goldstein et al., 2009;Rosenfeld et al., 2019). Such counteractive effect to anthropogenic emission reduction would
also slow down air quality improvement and reduce associated health benefits revealed by previous studies
(Markandya et al., 2018;Zhang et al., 2018).
Fire aerosol emissions and fire-induced land cover change manifest two major feedback mechanisms in climate-
fire-ecosystem interactions, showing synergistic or antagonistic effects at regional to global scales. These two distinct
feedback mechanisms compete with each other and increase the complexity of interactions among each interactive
component. It is noted that we only included the atmosphere and land modeling components of the CESM model to
investigate climate effects of global fires with other major components of the earth system including the ocean and
sea/land ice in the prescribed data mode. Enhanced climate sensitivity and feedback and uncertainties on a multi-
decadal scale might be expected in a fully coupled climate modeling system as previous studies revealed (Dunne et
al., 2012;Dunne et al., 2013;Hazeleger et al., 2010;Andrews et al., 2012). We suggest more comprehensive evaluations
at regional scales to investigate these complex interactions for major fire-prone regions. More advanced fire modeling
capabilities are also needed by integrating additional fire-related processes and climate effects such as fire emitted
brown carbon (Brown et al., 2018;Feng et al., 2013;Forrister et al., 2015;Liu et al., 2015;Wang et al., 2018;Zhang et
al., 2017; Zhang et al., 2019) and fire-vegetation-climate interactions and teleconnections (Garcia et al., 2016;Stark et
al., 2016). More evaluation metrics such as large wildfire extreme events should be considered in future studies to
improve our understanding of global and regional fire activities, their variations and trends, and their relationship with
decadal climate change.
**Code and data availability**
The Level-3 MODIS monthly AOD data from the Aqua platform (MYD08_M3,
http://dx.doi.org/10.5067/MODIS/MYD08_M3.006) used for model evaluation are available via NASA Level-1 and
Atmosphere Archive & Distribution System (LAADS) Distributed Active Archive Center (DAAC) in
https://ladsweb.modaps.eosdis.nasa.gov/missions-and-measurements/products/MYD08_M3/. The AERONET
Version 3 Level 2.0 AOT data are available at https://aeronet.gsfc.nasa.gov/. The GFED burned area and fire emission
datasets are available at http://www.globalfiredata.org/. All the CESM-RESFire model input and output data reported
in the paper are tabulated in the main text and archived on the Cheyenne high-performance computing system
(doi:10.5065/D6RX99HX) and High-Performance Storage System (HPSS) managed by the Computational &
Information Systems Lab (CISL) of NCAR. The modeling source code and data materials are available upon request,
which should be addressed to Yufei Zou (yufei.zou@pnnl.gov).
**Author contribution**
Y. Zou and Y. Wang designed the experiments and Y. Zou carried them out. Y. Zou developed the model code and
performed the simulations. Y. Zou and Y. Wang wrote the manuscript and all co-authors reviewed and edited the
manuscript.
**Competing interests**
The authors declare that they have no conflict of interest.
**Acknowledgments**
This work was supported by the National Science Foundation (NSF) through grant 1243220 and by the U.S.
Department of Energy (DOE)'s office of Science as part of the Regional and Global Climate Modeling Program (NSF-
DOE-USDA EaSM2). The Pacific Northwest National Laboratory (PNNL) is operated for DOE by Battelle Memorial
Institute under contract DE-AC05-76RL01830. H. Tian was supported by the NSF through grant 1243232. It has not
been subjected to any NSF review and therefore does not necessarily reflect the views of the Foundation, and no
official endorsement should be inferred.
We would like to acknowledge high-performance computing support from Cheyenne (doi:10.5065/D6RX99HX)
provided by NCAR's CISL, sponsored by the National Science Foundation. We are thankful to Steve Platnick for

processing the MODIS AOD data. We thank all the GFED team members for providing the GFED data at http://www.globalfiredata.org/. We thank Wei Min Hao, Brent Holben, Paulo Artaxo, Mikhail Panchenko, Sergey Sakerin, Rachel T. Pinker and their staff for establishing and maintaining the six AERONET sites used in this study. We thank Chandan Sarangi for the helpful discussion to improve the presentation of this work.

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

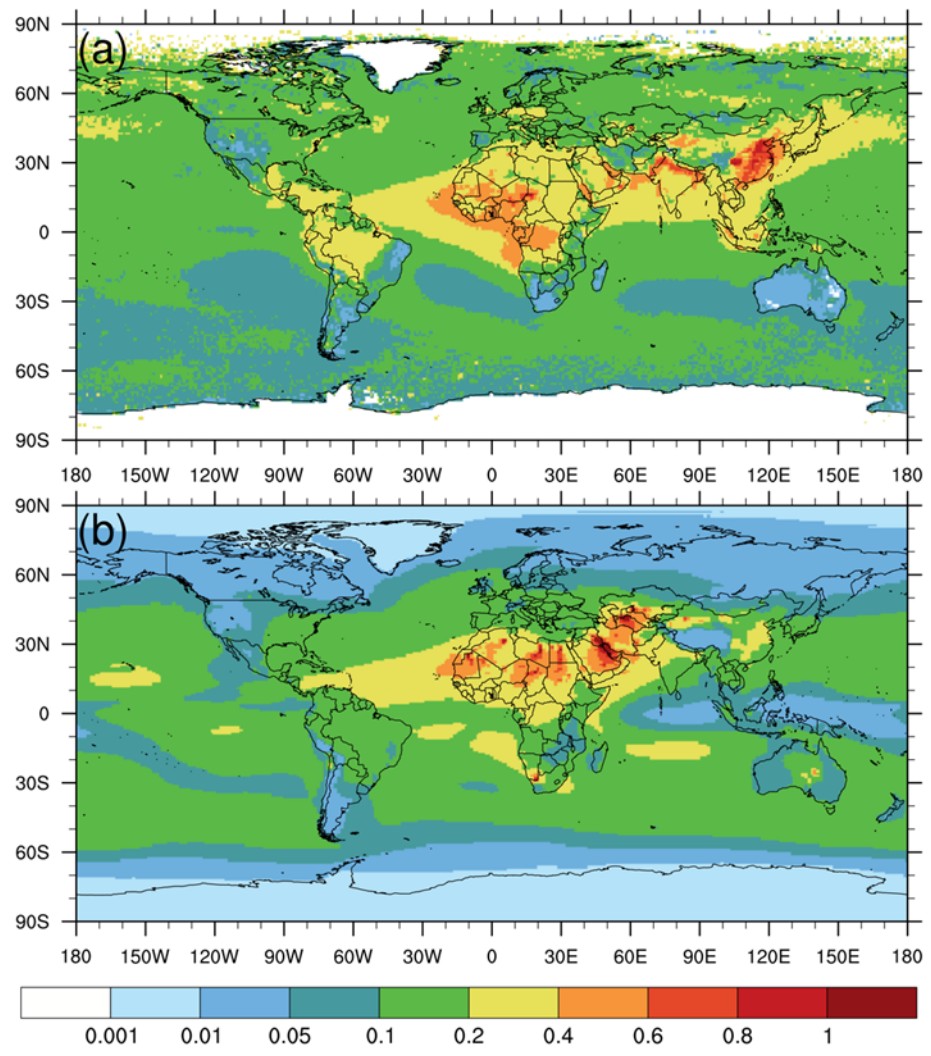


**Figure 1: Comparison of annual averaged column AOD at 550 nm from (a) MODIS aboard the Aqua satellite (2003-2010);**
**(b) CAM5 simulation averaged from 2001 to 2010.**

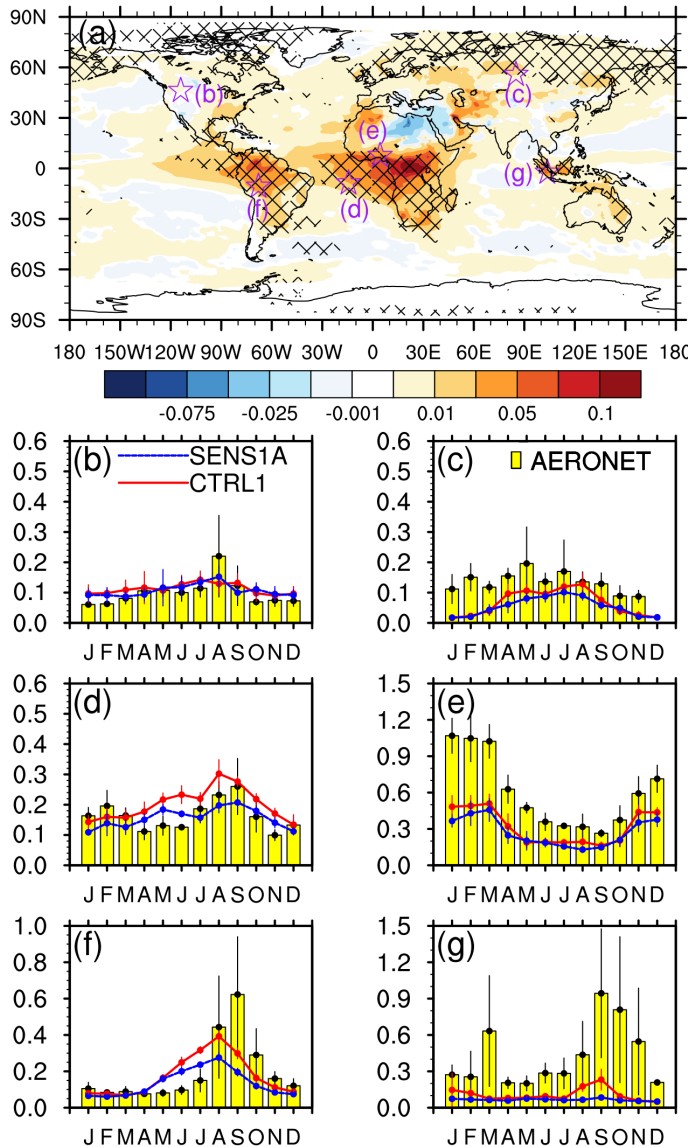


**Figure 2: CESM-RESFire simulation of (a) annual averaged fire contributed AOD at 550 nm (shading) in the present-day scenario (CTRL1-SENS1A). The stars denote the AERONET site location and the net meshes denote the 0.05 significance level of the two-tailed Student's t-test; (b) comparison with AERONET monthly AOT observations at 550 nm in Missoula (114.1°W, 46.9°N) during the 2000s. The error bars denote ±1 standard deviations of interannual variations in the simulations and observations, respectively.; (c) same as (b) but in Tomsk (85.1°E, 56.5°N); (d) same as (b) but in Ascension island (14.4°W, 8.0°S); (e) same as (b) but in Ilorin (4.3°E, 8.3°N); (f) same as (b) but in Rio Branco (67.9°W, 10.0°S); (g) same as (b) but in Jambi (103.6°E, 1.6°S).**

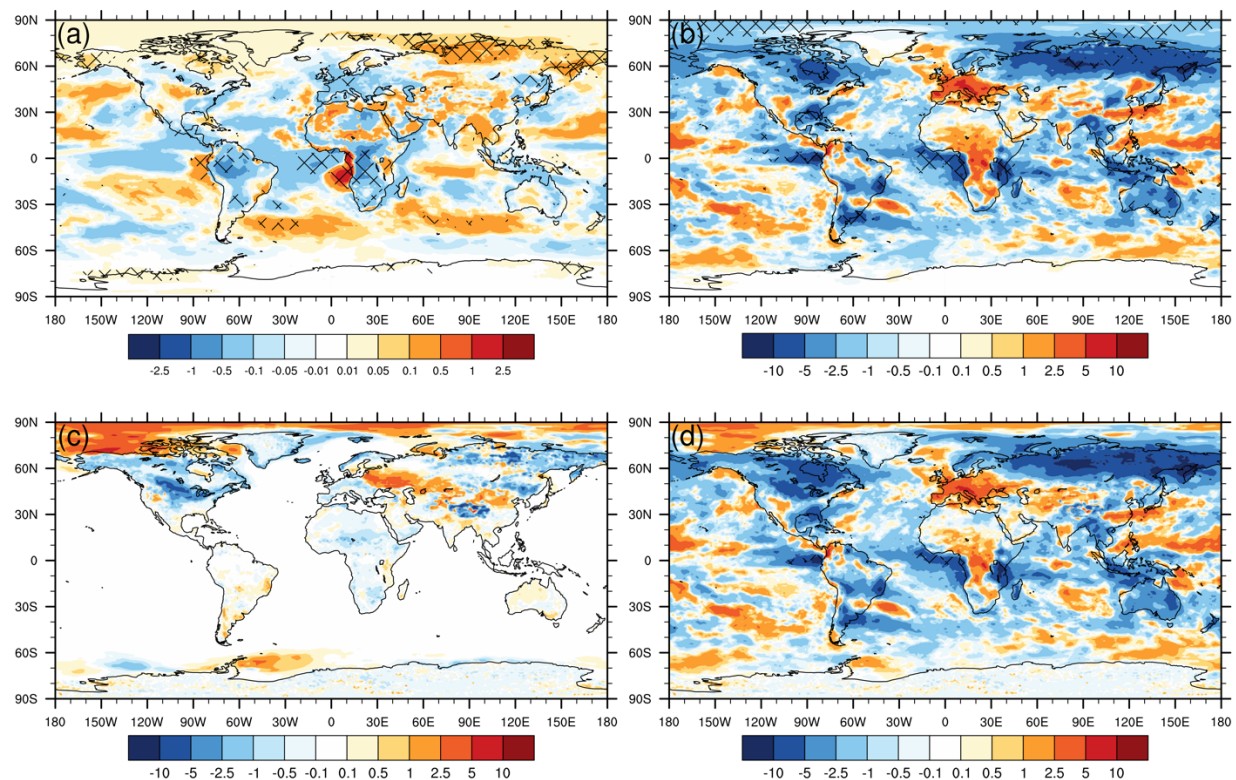

**Figure 3: Present-day simulation of fire contributed annual averaged radiative effects through (a) aerosol-radiation interactions (RE$_{ari}$ , W m$^{-2}$); (b) aerosol-cloud interactions (RE$_{aci}$ , W m$^{-2}$); (c) fire aerosol-induced surface albedo change (RE$_{sac}$ , W m$^{-2}$); (d) fire aerosol-related net radiative effects (RE$_{aer}$ , W m$^{-2}$). All these radiative effects are estimated as changes in the shortwave radiative flux at the TOA between CTRL1 and SENS1A experiments. The net meshes denote the 0.05 significance level.**

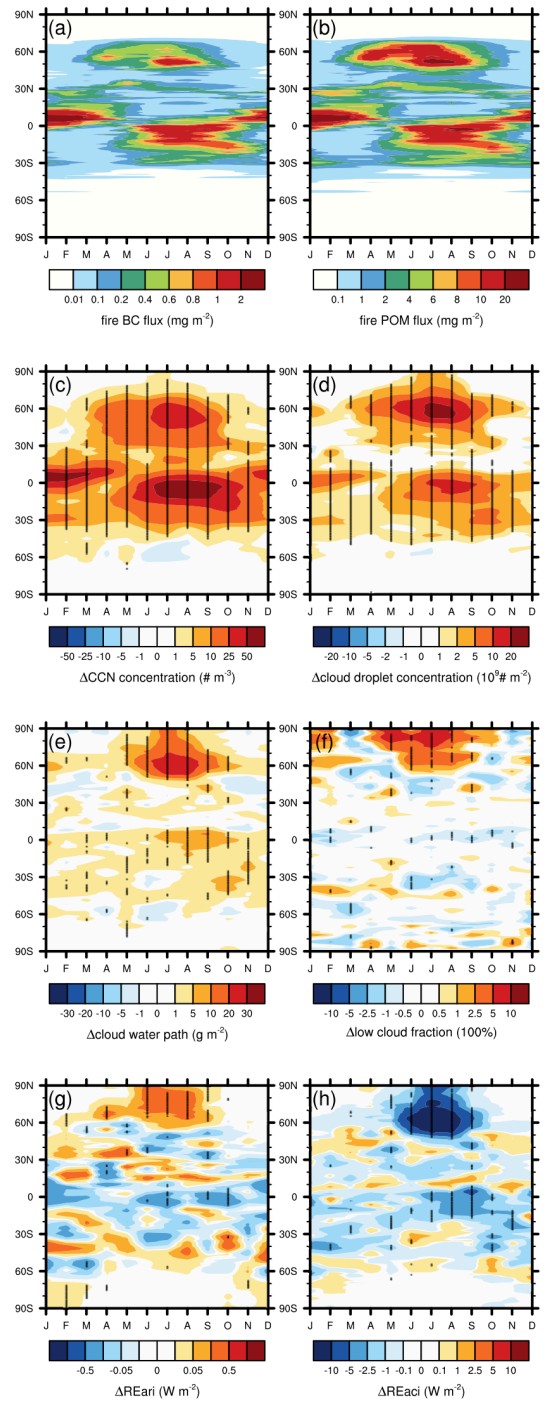


**Figure 4: Present-day simulation of zonal averaged time-latitude cross sections of (a) monthly BC fire emission fluxes (mg**
**m⁻²) in CTRL1; (b) monthly POM fire emission fluxes (mg m⁻²) in CTRL1; (c) fire-induced low-level (averaged below 800**
**hPa) cloud condensation nuclei (CCN, # m⁻³) concentration changes (CTRL1-SENS1A); (d) vertically-integrated cloud**
**droplet number concentration (CDNUMC, 10⁹# m⁻²) changes (CTRL1-SENS1A); (e) cloud water path (CWP, g m⁻²) changes**
**(CTRL1-SENS1A); (f) low cloud cover fraction (100%) changes (CTRL1-SENS1A); (g) radiative effect changes (CTRL1-**
**SENS1A) by fire aerosol-radiation interactions (RE$_{ari}$ , W m⁻²); (h) radiative effect changes (CTRL1-SENS1A) by fire**
**aerosol-cloud interactions (RE$_{aci}$ , W m⁻²). The dots in (c)-(h) denote the 0.05 significance level.**

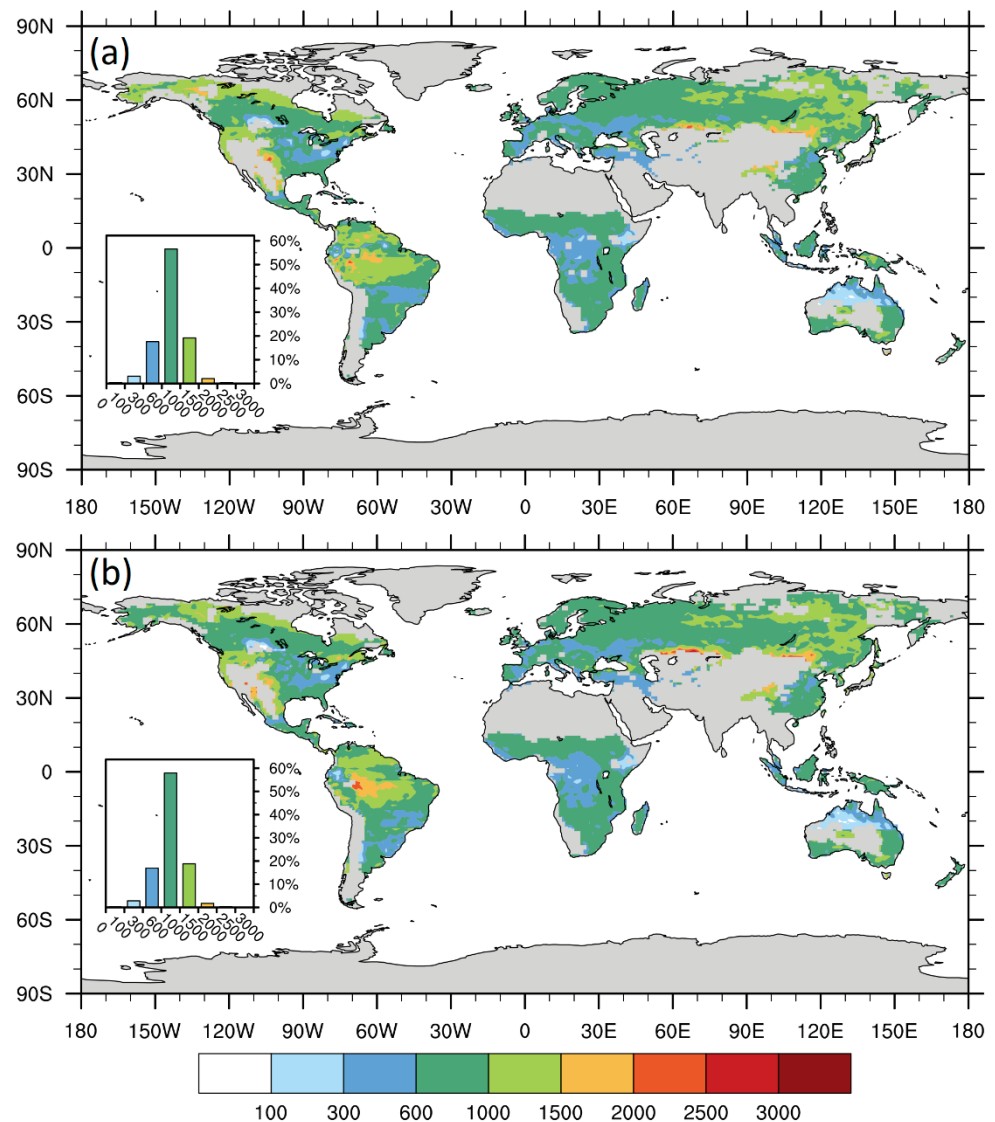


Figure 5: Comparison of CESM-RESFire simulated annual median injection heights (m) of fire plumes in the (a) present-day (CTRL1) and (b) RCP4.5 (CTRL2) scenarios. The inlets show statistical distributions of all plume injection heights in model grid cells of each scenario.

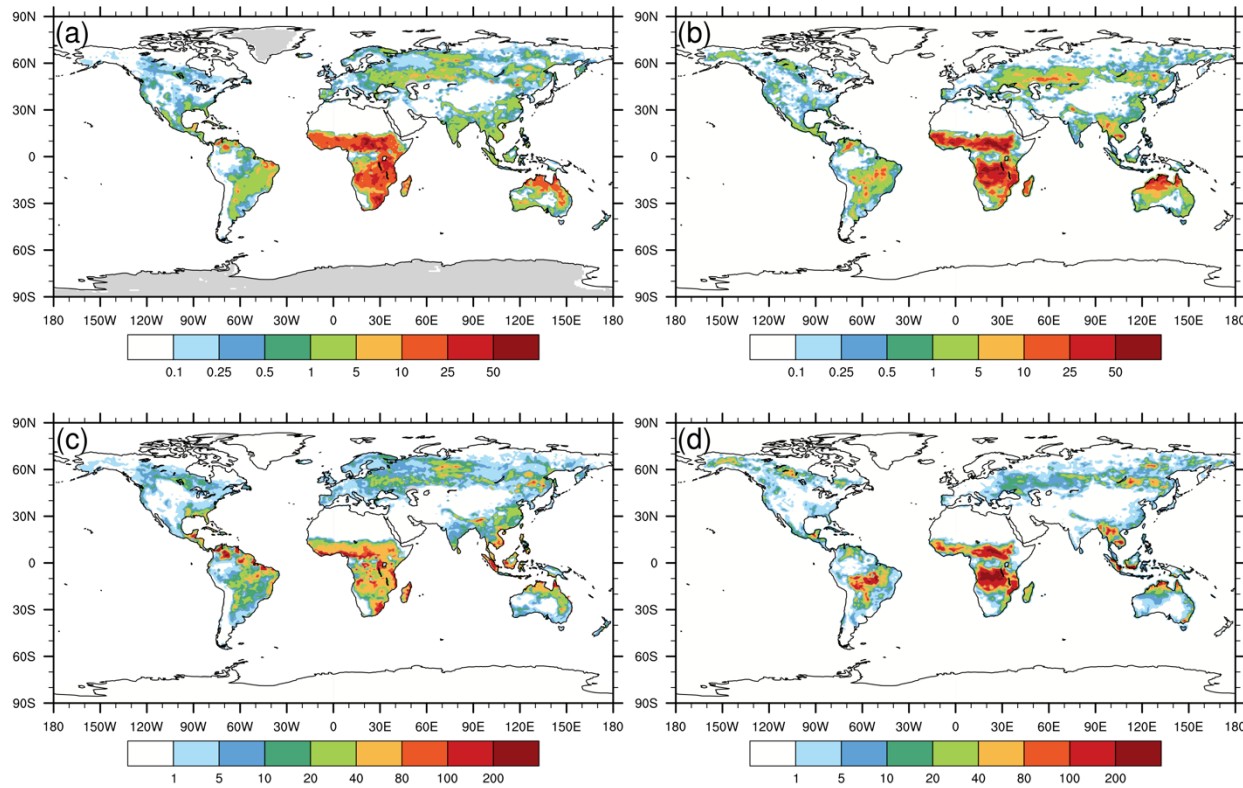


**Figure 6: Comparison of CESM-RESFire simulations and GFED4.1s data. (a) ensemble averaged annual fractional burned**
**area (% yr⁻¹) simulation; (b) 10-year averaged (2001-2010) annual fractional burned area (% yr⁻¹) based on the GFED4.1s**
**data; (c) ensemble averaged annual fire carbon emission (gC m⁻² yr⁻¹) simulation; (d) 10-year averaged (2001-2010) annual**
**fire carbon emission (gC m⁻² yr⁻¹) based on the GFED4.1s data.**

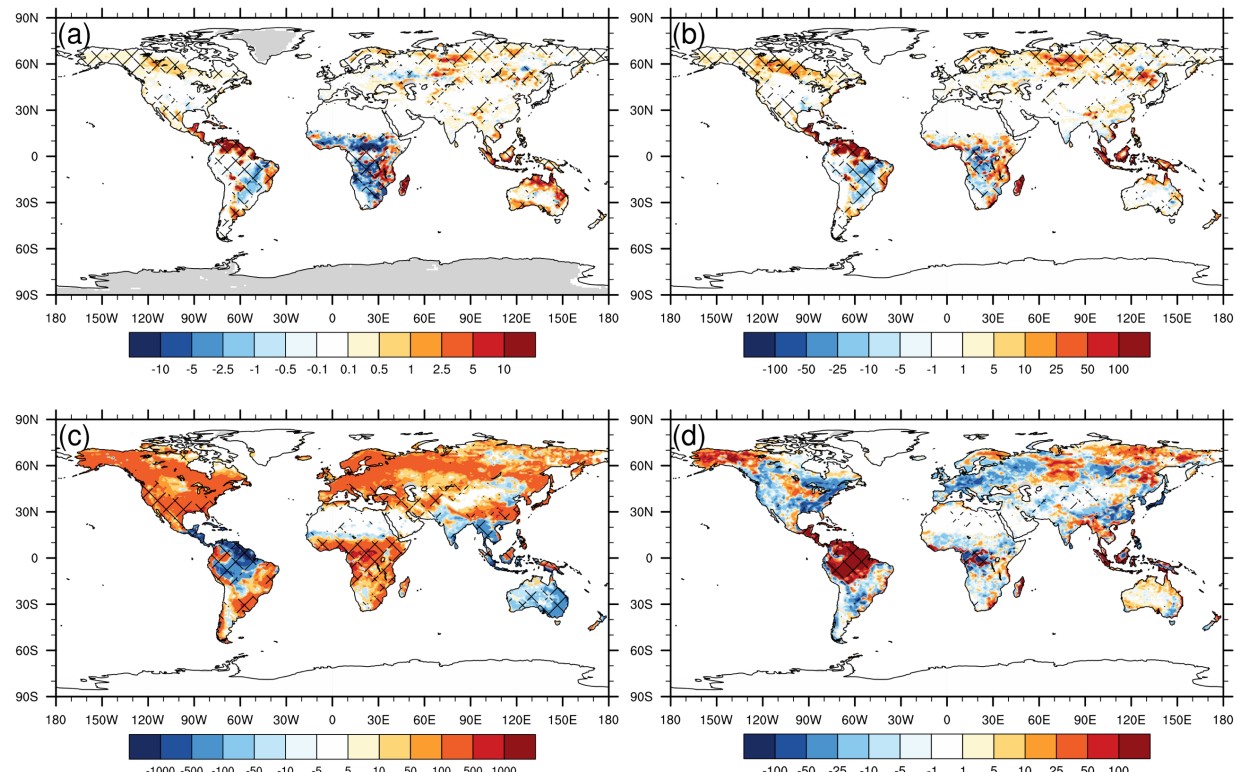

Figure 7: CESM-RESFire simulated changes between the RCP4.5 future scenario and the present-day scenario (CTRL2-CTRL1) in (a) annual fractional burned area (% yr$^{-1}$); (b) annual averaged fire carbon emissions (gC m$^{-2}$ yr$^{-1}$); (c) annual averaged GPP (gC m$^{-2}$ yr$^{-1}$); (d) annual averaged NEE (gC m$^{-2}$ yr$^{-1}$). The net meshes denote the 0.05 significance level.

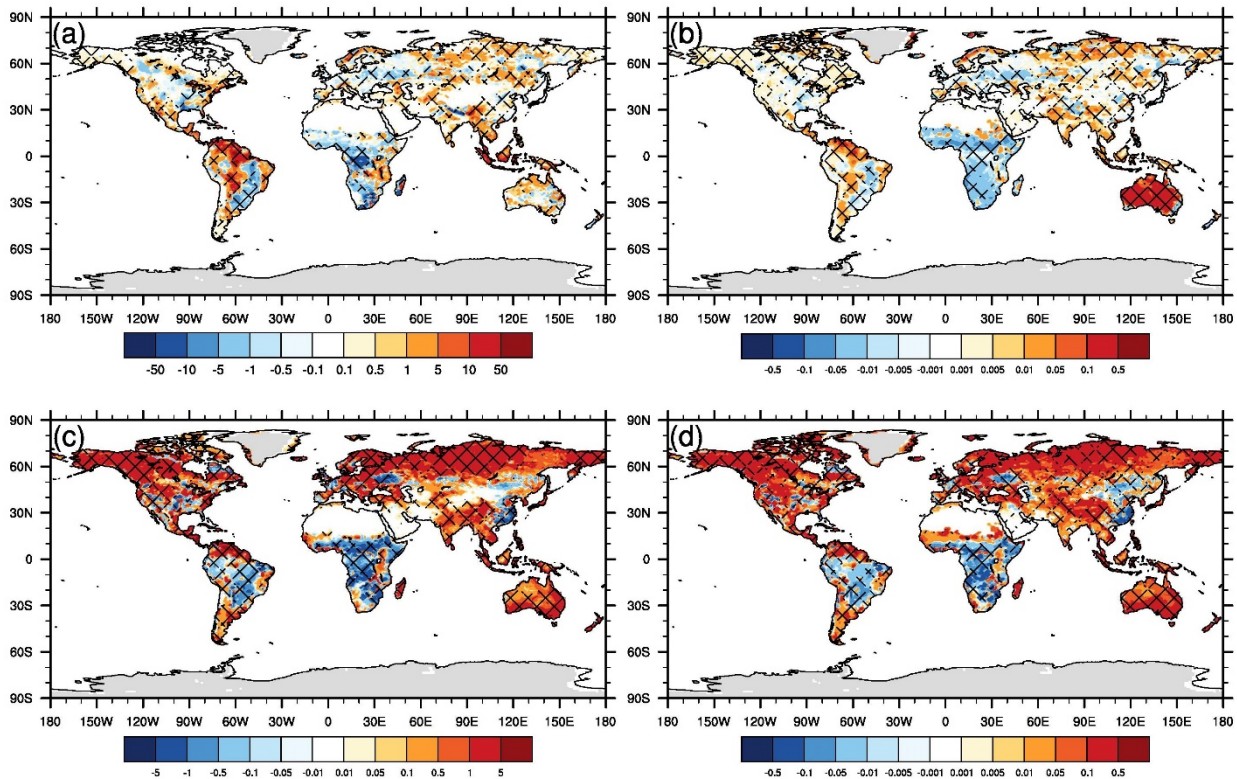


**Figure 8: CESM-RESFire simulated changes in fire-related variables between the RCP4.5 future scenario and the present-**
**day scenario (CTRL2-CTRL1). (a) changes in annual total fire ignition (NFIRE, 1E-3 count km$^{-2}$ yr$^{-1}$); (b) changes in annual**
**averaged fire combustion factors (FCF, unitless); (c) changes in annual averaged fire spread rates (FSR_DW, cm s$^{-1}$); (d)**
**changes in annual averaged fire spread factors (FSF, unitless). The net meshes denote the 0.05 significance level.**

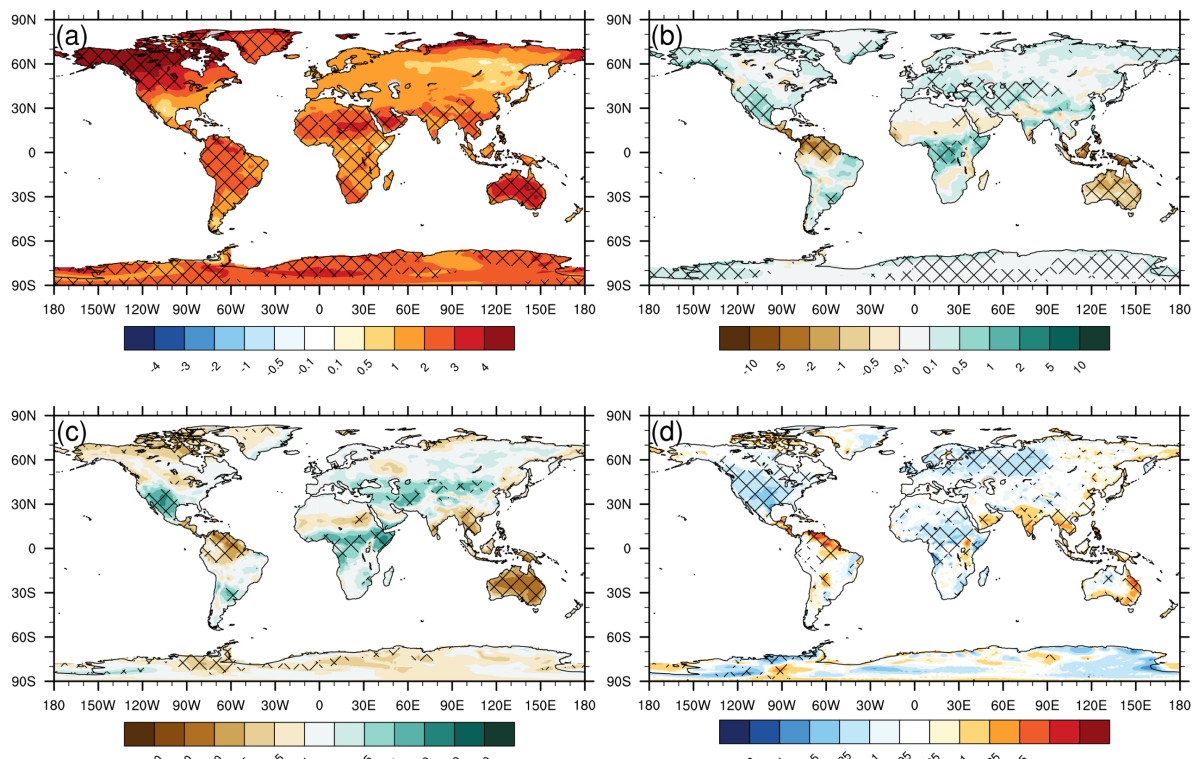


**Figure 9: CESM-RESFire simulated changes in fire weather variables between the RCP4.5 future scenario and the present-day scenario (CTRL2-CTRL1). (a) changes in surface temperature (K); (b) changes in total precipitation rate (mm day⁻¹); (c) changes in surface relative humidity (%); (d) changes in surface wind speed (m s⁻¹). The net meshes denote the 0.05 significance level. For clear comparison with fire changes in Fig. 7 and 8, only fire weather changes over land are shown.**

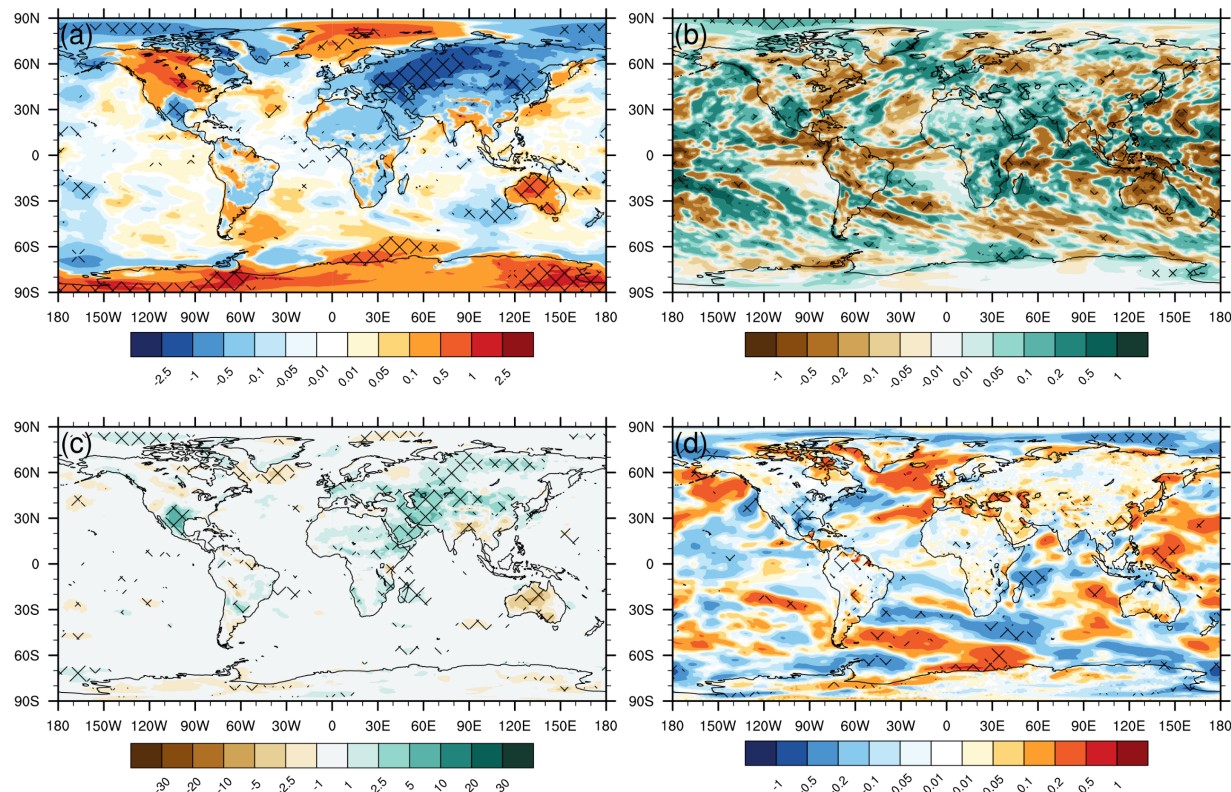


**Figure 10: Fire induced changes in fire weather variables between the RCP4.5 future scenario and the present-day scenario**
**((CTRL2-CTRL1)-(SENS2B-SENS1B)). (a) fire induced changes in surface temperature (K); (b) fire induced changes in**
**total precipitation rate (mm day$^{-1}$); (c) fire induced changes in surface relative humidity (%); (d) fire induced changes in**
**surface wind speed (m s$^{-1}$). The net meshes denote the significance level of $p$=0.05.**

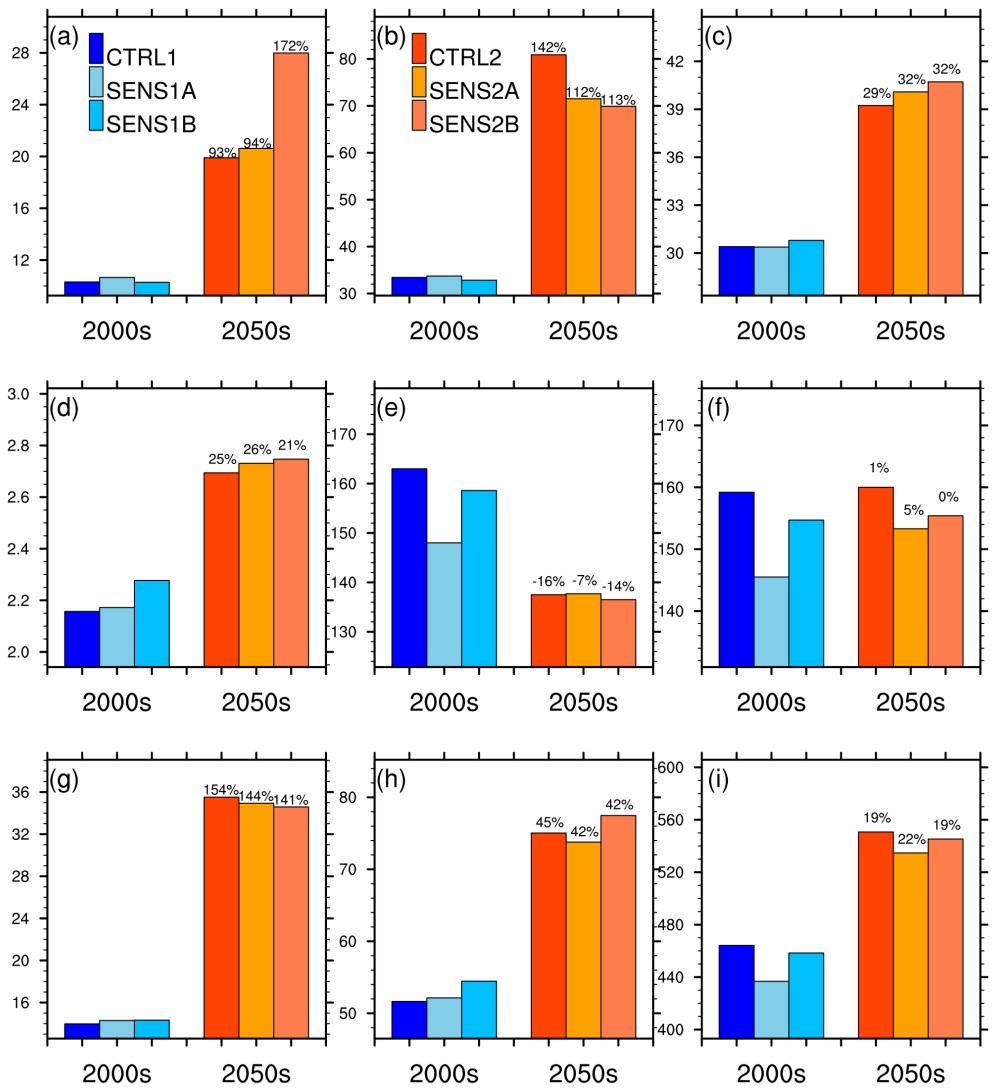

**Figure 11: Comparison of annual burned area (Mha yr⁻¹) in each region among different time periods and sensitivity**
**experiments. (a) North America; (b) South America; (c) Eurasia excluding Middle East and South Asia; (d) Middle East**
**and North Africa; (e) Northern Hemisphere Africa; (f) Southern Hemisphere Africa; (g) South and Southeast Asia; (h)**
**Oceania; (i) global total BA. The percentage numbers above projection columns are changes of burned area in the 2050s**
**relative to their counterpart experiments in the 2000s. The spatial distributions of these regions are shown in Fig. S4 of the**
**Supplement.**

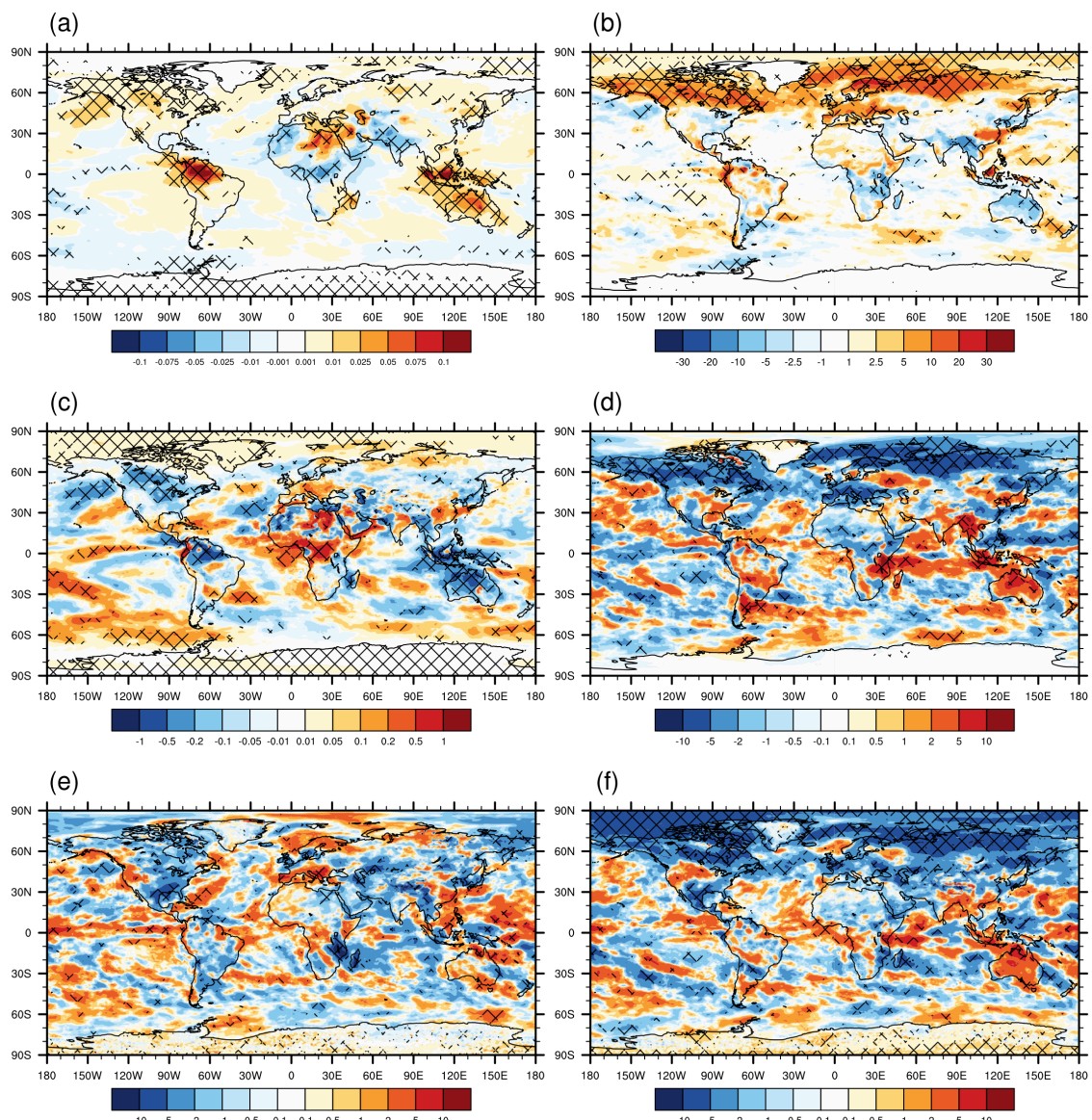


**Figure 12: Changes in fire induced weather conditions and climate radiative forcing between the RCP4.5 future scenario**

**and the present-day scenario. (a) changes in annual averaged column AOD at 550 nm (unitless, (CTRL2-SENS2A)-**

**(CTRL1-SENS1A)); (b) changes in cloud liquid water path (g m$^{-2}$, (CTRL2-SENS2A)- (CTRL1-SENS1A)); (c) changes in**

**RE$_{ari}$(W m$^{-2}$, (CTRL2-SENS2A)- (CTRL1-SENS1A)); (d) changes in RE$_{aci}$(W m$^{-2}$, (CTRL2-SENS2A)- (CTRL1-SENS1A));**

**(e) changes in RE$_{lcc}$ (W m$^{-2}$, (SENS2A-SENS2B)- (SENS1A-SENS1B)); (f) changes in RE$_{fire}$ (W m$^{-2}$, (CTRL2-SENS2B)-**

**(CTRL1-SENS1B)). The net meshes denote the 0.05 significance level.**

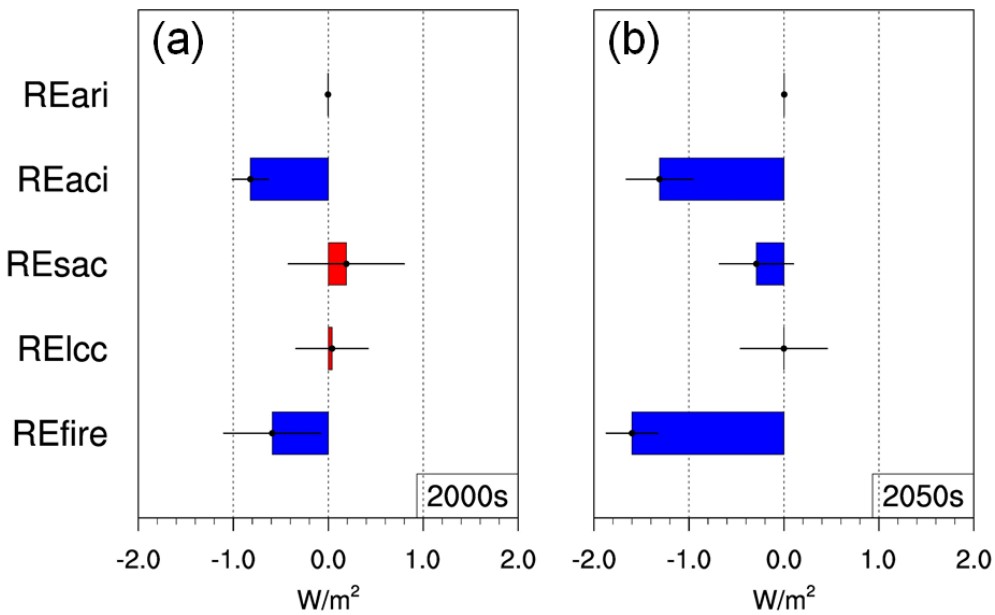


**Figure 13: Comparison of CESM-RESFire simulated fire radiative effects (W m⁻²) in (a) the present-day scenario and (b)**
**the RCP4.5 future scenario. The error bars denote standard deviations of interannual variations during each 10-year**
**simulation period.**

**Table 1: Fire sensitivity simulation experiments for the present-day and RCP4.5 future scenarios**

| Scenario | Present-day (2000) | | | Future (RCP4.5) | | |
|---|---|---|---|---|---|---|
| Name | CTRL1 | SENS1A | SENS1B | CTRL2 | SENS2A | SENS2B |
| Time | 2001-2010 | 2001-2010 | 2001-2010 | 2051-2060 | 2051-2060 | 2051-2060 |
| Atmosphere | CAM5 | CAM5 | CAM5 | CAM5 | CAM5 | CAM5 |
| Land | CLM4.5 | CLM4.5 | CLM4.5 | CLM4.5 | CLM4.5 | CLM4.5 |
| Ocean | Climatology | Climatology | Climatology | RCP4.5 data | RCP4.5 data | RCP4.5 data |
| Sea ice | Climatology | Climatology | Climatology | RCP4.5 data | RCP4.5 data | RCP4.5 data |
| Non-fire emissions | IPCC AR5 emission data | IPCC AR5 emission data | IPCC AR5 emission data | RCP4.5 data | RCP4.5 data | RCP4.5 data |
| Fire emissions | Online fire aerosols with plume rise | — | — | Online fire aerosols with plume rise | — | — |
| Land cover | Fire disturbance on present-day conditions | Fire disturbance on present-day conditions | Fixed present-day conditions in 2000 | Fire disturbance on RCP4.5 conditions | Fire disturbance on RCP4.5 conditions | Fixed RCP4.5 conditions in 2050 |


**Table 2: Comparison of fire-related radiative effects in the present-day (CTRL1-SENS1A) and RCP4.5 future (CTRL2-**
**SENS2A) scenarios based on this work and previous studies**

| Unit: W m$^{-2}$ | This work | | Jiang et al. (2016) | Ward et al. (2012) | |
|---|---|---|---|---|---|
| Time | 2000s | 2050s | 2000s | 2000s (CLM3/GFEDv2) | 2100s (CCSM/ECHAM) |
| RE$_{ari}$ | -0.003$\pm$0.013[*] | 0.003$\pm$0.033 | 0.16$\pm$0.01 | 0.10/0.13 | 0.12/0.25 |
| RE$_{aci}$ | -0.82$\pm$0.19 | -1.31$\pm$0.35 | -0.70$\pm$0.05 | -1.00/-1.64 | -1.42/-1.74 |
| RE$_{sac}$ | 0.19$\pm$0.61 | -0.29$\pm$0.39 | 0.03$\pm$0.10 | 0.00/0.01 | 0.00/0.00 |
| RE$_{aer}$ | -0.64$\pm$0.48 | -1.59$\pm$0.33 | -0.55$\pm$0.07 | -0.90/-1.50 | -1.30/-1.49 |
| RE$_{lcc}$ | 0.04$\pm$0.38 | -0.006$\pm$0.457 | — | -0.20/-0.11 | -0.23/-0.29 |
| RE$_{fire}$ | -0.59$\pm$0.51 | -1.60$\pm$0.27 | -0.55$\pm$0.07 | -0.55[**]/— | -0.83/-0.87[**] |

*: the numbers after $\pm$ denote standard deviations of interannual variations;
**: the net radiative forcing includes other effects such as GHGs and climate-BGC feedback;

**Table 3. Comparison of fire and carbon budget variables between CESM-RESFire simulations and previous studies and**
**benchmarks**

| Variables | Time Period | This work | | CLM-LL2013 (Li et al., 2014) | Benchmark | Sources |
|---|---|---|---|---|---|---|
| Models | | RESFire-CRUNCEP | RESFire-CAM5c | CLM4.5-DATM | | |
| Burned area (Mha yr$^{-1}$) | 1997-2004 | 508 ± 15 | 472 ± 14 | 322 | 510 ± 27 | GFED4.1s (Giglio et al., 2013; Randerson et al., 2012) |
| Fire carbon emissions (Pg C yr$^{-1}$) | 1997-2004 | 2.3 ± 0.2 | 2.6 ± 0.1 | 2.1 | 2.2 ± 0.4 | GFED4.1s (van der Werf et al., 2017) |
| NEE (Pg C yr$^{-1}$) | 1990s | -2.6 ± 0.6 | -2.0 ± 1.3 | -0.8 | -1.1 ± 0.9 -2.0 ± 0.8 | IPCC AR5 (Ciais et al., 2013) 10 models average (Piao et al., 2013) |
| GPP (Pg C yr$^{-1}$) | 2000-2004 | 142 ± 2 | 142 ± 1 | 130 | 133 ± 15 | 10 models average (Piao et al., 2013) |
| NPP (Pg C yr$^{-1}$) | 2000-2004 | 62 ± 1 | 63 ± 0.7 | 54 | 54 | Zhao and Running (2010) |


**Table 4. Comparison of carbon budget variables between the CRUNCEP data atmosphere driven fire simulations based on**
**CESM-RESFire and CLM-LL2013**

| Variables | CESM-RESFire | | | CLM-LL2013 (Li et al., 2014) | | |
|---|---|---|---|---|---|---|
| Unit: Pg C yr$^{-1}$ | $\Delta$Fire | Fire on | Fire off | $\Delta$Fire | Fire on | Fire off |
| NEE | 1.58 | -2.67 | -4.25 | 1.0 | -0.1 | -1.1 |
| $C_{fe}$ | 2.08 | 2.08 | 0.0 | 1.9 | 1.9 | 0.0 |
| -NEP+$C_{lh}$ | -0.5 | -4.75 | -4.25 | -0.9 | -2.0 | -1.1 |
| NEP | 0.5 | 4.8 | 4.3 | 0.8 | 3.0 | 2.3 |
| NPP | 0.4 | 61.7 | 61.3 | -1.9 | 49.6 | 51.6 |
| Rh | -0.1 | 56.9 | 57.0 | -2.7 | 46.6 | 49.3 |
| GPP | -0.1 | 142.3 | 142.4 | -5.0 | 118.9 | 123.9 |
| Ra | -0.5 | 80.6 | 81.1 | -3.1 | 69.3 | 72.4 |
| $C_{lh}$ | 0.0 | 0.05 | 0.05 | -0.1 | 1.0 | 1.1 |


**Table 5. Comparison of carbon budget variables between CESM-RESFire sensitivity experiments and previous studies**

| Variables | This work | | | | | | Kloster et al. (2010) Kloster et al. (2012) | |
|---|---|---|---|---|---|---|---|---|
| Time (scenario) | 2000s (CTRL1) | 2050s (CTRL2) | 2000s (SENS1A) | 2050s (SENS2A) | 2000s (SENS1B) | 2050s (SENS2B) | 2000s | 2050s |
| Burned area (Mha yr$^{-1}$) | 464±19 | 551±16 (↑19%)* | 437±17 (↓6%)** | 535±19 (↓3%) | 458±18 (↓1%) | 545±18 (↓1%) | 176-330 | — |
| Fire carbon emissions (Pg C yr$^{-1}$) | 2.5±0.1 | 5.0±0.3 (↑100%) | — | — | — | — | 2.0-2.4 | 2.7/ 3.4 |
| GPP (Pg C yr$^{-1}$) | 141±1.2 | 146±1.1 (↑4%) | 143±1.0 (↑1%) | 149±1.3 (↑2%) | 142±1.5 (↑1%) | 150±1.3 (↑3%) | — | — |
| NEP (Pg C yr$^{-1}$) | 1.4±0.04 | 1.5±0.04 (↑7%) | 1.4±0.04 (→0%) | 1.6±0.04 (↑7%) | 1.4±0.02 (→0%) | 1.6±0.05 (↑7%) | — | — |
| NEE (Pg C yr$^{-1}$) | 1.2±0.03 | 1.6±0.05 (↑33%) | 1.2±0.02 (→0%) | 1.6±0.05 (→0%) | 1.2±0.02 (→0%) | 1.6±0.05 (→0%) | — | — |

*: percentage numbers in the parentheses under CTRL2 denote relative changes comparing with the CTRL1
scenario.
**: percentage numbers in the parentheses under SENSx (x=1 or 2) denote relative changes comparing with the
corresponding CTRLx (x=1 or 2) scenarios.