# Peer review of "Using CESM-RESFire to Understand Climate-Fire-Ecosystem"

_Atmospheric Chemistry and Physics, 2019_

## Referee Comment (RC1) · Anonymous Referee #2 · 17 Sep 2019

The manuscript by Zou et al. presents an analysis of the interactions between climate, wildfires, ecosystems, and radiative balance in a recently (further) developed modelling system, CESM-RESFire. The methodology includes a suitable set of sensitivity experiments that provide substantial new insight into the role of different types of potential interactions (mainly aerosol effects and land cover changes) in driving present-day radiative effects of wildfires, and their future radiative forcing. It features some novel aspects compared to previous studies, especially when it comes to the types of feedbacks allowed and investigated, and provides a useful contribution to the improvement of our poor understanding of the role of fire in the Earth system. The manuscript is nicely written, and well within the scope of Atmospheric Chemistry and Physics. I find

it worthy of publication, following some (mostly minor) improvements that I describe below.

GENERAL COMMENTS:

- The title, abstract and conclusions (as well as the main text) leave the reader thinking that the full climate effects of wildfires are examined in the current study. However, this is somewhat misleading, as full climate responses (i.e. temperature, precipitation, humidity etc changes) are not explored or discussed, even if they are partially included (I am saying "partially" since the oceans and sea-ice are fixed). The study goes up to radiative effect and radiative forcing quantification, and that should be reflected more accurately in the different parts of the text. In my specific comments below, there are some suggestions for amending this, but the authors should make an effort to do so further throughout the text.

- In addition, the future radiative impacts (whose study presumably is a core aim of this work) are discussed very briefly towards the end of Sect. 3.3, and in a way that does not seem accurate/consistent with what is shown on the maps (see related comment below).

- The past tense is often used in the text to refer to the work presented here, where the present tense would be more appropriate/standard. For example "We provided a brief model description and sensitivity experiment settings in Section 2...", where "provide" would probably read better. I suggest making this amendment to wherever applicable in the text.

SPECIFIC COMMENTS:

Page 2, Lines 1-3: I suggest changing the title to "Using CESM-RESFire to Understand Climate-Fire-Ecosystem Interactions and their Implications for Radiative Forcing". The title as it stands currently is misleading, as "implications for decadal climate variability" were not examined at all in this study. Generally speaking, it is radiative forcing/effects

that were examined, rather than climate (temperature, humidity, precipitation etc) effects.

Page 1, Line 20: For the same reason, I suggest rephrasing to "...and their impacts on fire activity and radiative forcing".

Page 2, Line 38: Please add ", respectively" at the end of the sentence.

Page 2, Lines 57-58: "are further confounded by natural processes and human interferences" – human and natural processes have been mentioned in the previous sentence. Why repeat them?

Page 2, Line 69: "used the same approach" – suggest changing to "used the same unidirectional approach".

Page 2, Line 71: The term "fixed" may not be fully accurate here. For example, to my knowledge, Tosca et al. (2013) performed simulations with and without aerosol emissions, with no "fixing" per se involved.

Page 3, Line 79: I do not think "feedback in" is needed.

Page 3, Lines 81-84: Change past tense to present tense (just another example).

Sect. 2.1: Gas-phase chemistry (e.g. ozone and its precursors) is not mentioned at all in the model description – or anywhere really. If such a mechanism is not included, this should be mentioned (along with acknowledging the potentially sizeable effect of this missing process), and if included, the authors should describe in what fashion it is included.

Sect. 2.1: No mention at all of biogenic aerosols.

Page 3, Line 97: Probably "microphysics" and not "macrophysics"?

Page 4, Line 110: Please mention the year for the future scenario. It's mentioned later, but worth mentioning it here too.

Page 4, Lines 138-140: Suggested rephrasing – "...we allowed the semi-static historical LCC data for the year 2000 from the version 1 of the Land-Use History A product (LUHa.v1) (Hurtt et al., 2006) to be affected by post-fire vegetation changes (Zou et al., 2019)".

Page 5, Lines 150-151: "given great uncertainties in future projections of these inputs" - Are these uncertainties larger than for the rest of the variables considered here?

Page 5, Lines 161-164: Yes, but are the timescales long enough in this case for this assumption to hold? Please discuss.

Page 5, Line 175: "the Ghan's method" -> "the Ghan method"

Equations (1): The way these equations are written is very confusing. First of all because of the dashes ("-") and the minuses appearing identical, and also because of the use of column (:). I suggest the following format:

"RE of interaction of radiation with fire aerosol: RE = $\Delta$(F -F)"

(with the appropriate subscripts in each case)

Page 6, Line 188: "nonnegligible" -> "non-negligible"

Page 6, Line 200: "budge" -> "budget"

Page 7, Lines 220-221: "However, the model well captured the high AOD regions over the Northern and Southern Hemispheres of Africa" – I am not sure I see this on Fig. 1. Therefore the statement seems too confidently positive.

Page 7, Line 228: The AERONET measurements cannot be characterised as "in situ". They are also remotely sensed.

Fig. 3: Please specify that this is TOA radiative effect.

Page 7, Lines 243-247: There are some areas that experience pronounced positive forcing due to fire aerosol-cloud interactions. The most prominent ones are Europe

and most of Africa. Presumably that is because of black carbon stabilisation effects? But why would these be more important in these specific regions? Any thoughts? Please comment.

Page 8, Lines 248-249: Why are there areas with both positive and negative changes? Why is Africa pretty much all negative? These are interesting features. Please elaborate.

Page 8, Lines 262-263: Please specify that the Jiang et al. (2016) study was performed with the same atmospheric model as in the current study (though older version?), as it is useful for the reader to know.

Page 9, Line 293: I am not sure where the +51% value comes from. From Table 2, the Raci is -1.31 for 2050 in this study and -1.42 in the CCSM study. Or do the authors mean something different and I am missing the point? In any case, I think it should be made clearer where the +51% value comes from.

Page 9, Lines 313-315: "Such difference is also consistent with the changes in different versions of the GFED datasets, which show a 11% increase of global fire carbon emissions in the latest GFED4s as compared with the old GFED3 for the overlapping 1997-2011 time period (van der Werf et al., 2017)" – Do the authors mean that there is an upward "trend" between older and newer GFED emissions versions, implying that eventually the GFED emissions will match the online model? That's a rather simplistic reasoning and needs to be supported further or phrased differently.

Page 11, Lines 361-363: "Though we mainly focused on fire-climate interactions without consideration of human impacts in this study, the RESFire model is capable of reproducing the anthropogenic interference on fire activity as observed from the space (Zou et al., 2019)" – This needs some more explanation. The common understanding is that in Northern Hemisphere Africa the decline in burned area is due to agricultural conversion and resulting landscape fragmentation (e.g. Andela et al., 2017). Is this a process that is represented in this particular model? Please clarify and discuss.

Page 11, Lines 390-394: The evidence to support this statement is somewhat weak. First of all, the precipitation changes (Fig. 9c) are not significant almost everywhere (therefore, not much difference in that respect to the wind changes). Secondly, the match between locations with decreased precipitation and increased burned area (and the other way around) is not always clear (e.g. the north of Siberia experiences increases in burned area, but simultaneous increases in precipitation; there are other examples too). It would be best to discuss this in a more quantitative fashion, e.g. report the spatial correlation coefficients between burned area and driver variables to extract more robust conclusions?

Figure 11: This figure is not really discussed in any insightful way, beyond just stating that such effects "might compensate biogeochemical warming effects of deforestation related carbon-cycle changes". How does each individual variable shown affect warming/cooling patterns, and which of these variables seems to be more important, based on this analysis?

Page 12, Lines 427-431: This discussion is rushed and I am not sure I follow the reasoning. It is stated that the radiative forcing of aerosol-radiation interactions "show similar patterns with Fig. 3a, with generally cooling effects over the vicinities of fire areas and warming effects over the downwind regions". Where do we see this? In Fig. 3a, this was evident e.g. in and around Africa (and possibly South America), but I cannot see this in Fig. 12c. Then for aerosol-cloud interactions, it is stated that there are "warming effects in Southeast Asia and Australia due to local cloud changes", but how are these features consistent with Fig. 3b, in which the inclusion of fire caused negative radiative effects due to aerosol-cloud interactions over those regions (as was the case for northern high latitudes).

Page 13, Line 447: Please add "fire" between "significant" and "aerosol".

Page 13, Lines 455-456: Please change "climate effects" to "radiative effects", as the former implies that effects on temperature, precipitation etc due to fires were also examined (which is not the case).

Page 13, Line 465: Please change "their" to "its".

---

## Referee Comment (RC2) · Anonymous Referee #1 · 2 Oct 2019

General comments: In the manuscript 'Understanding Climate-Fire-Ecosystem Interactions Using CESM-RESFire and Implications for Decadal Climate Variability', Zou et al. explored complex interactions between climate change, fire, and ecosystem using a global Earth System Model equipped with a coupled fire module. They estimated the global net radiative effects and NEE changes due to fire aerosols and fire-induced land cover changes under present-day and future scenarios. The topic is interesting and relevant to the scope of ACP. Overall, this is a nicely written manuscript with a clear description of data, model design and results. I recommend it to be published after some minor modifications suggested below.

[Figure]

Specific comments: My only major concern is the present manuscript lacks a detailed discussion about the uncertainty of the simulations and calculations. Specifically, although most current state-of-art fire models (including RESFire used in this study) may be able to reproduce the main spatial variability of fire emissions (and fire pollutants) under current climate condition, their ability to simulate temporal variability, as well as the changes under a changing climate has not been validated. As mentioned by the authors, some important processes (such as the lightning changes in the warming future) are also ignored in this study. It will be interesting to know how does it lead to changes in the simulated fire impacts in the future scenario. I believe this paper will be benefited from adding some discussions on this topic.

Minor and technical comments: Page 1, Line 17: "The complex climate-fire-ecosystem interactions were not included in previous climate model studies". I suggest softening the tune here. Some components of the interactions between climate, fire, and ecosystem have been considered in previous studies (although they were not necessarily incorporated into, or might not be represented thoroughly in a fully coupled online model).

Page 2, Line 58: "These processes were not included in previous climate model studies". Similar to the above, this sentence is way too assertive.

Page 3, Line 102-103: Since the new scheme is not implemented in this study (and the readers don't know the strength of the new approach), you don't have to mention it here. Removing this sentence won't affect the integrity of this paper.

Page 7, Line 218-220: In addition to biogenic organic aerosols, can an underestimation of fire emissions be another reason for low simulated aerosols?

Page 7, Line 246-247: Any physical explanation for the differences between the signs of aerosol-cloud interactions and aerosol-radiation interactions?

Page 8, Line 279: It would be good to briefly introduce this plume rise parameterization

(e.g., based on what measurements? Global universal or regional-based?)

Page 11, Line 376-379: The terms 'fire combustion factors', 'fire spread distribution', and 'fire spread factors' are probably not familiar to many readers. Please consider a short explanation on these parameters (i.e., what do they mean physically).

Page 11, Line 388-389: I don't quite understand the causal relationship stated in this sentence. The changes in wind speed are higher over the ocean than that over land, but this could be simply due to the larger magnitude of wind speed over the ocean. Relatively smaller changes in land wind speed could still have large impacts on fire spread and burned area.

Page 25, Figure 2: Please align tick label '0.1' with other tick labels in panels b, c, d.

Page 27, Line 817: Should the unit of CDNUMC '10ˆ9 # /m2' (as correctly shown in panel d)?

Page 30, Figure 7: The colors in panel c don't have enough separation. Please use another scale.

Page 32, Figure 9: If my understanding is correct, the data in this figure show the differences of fire modifications on weather variables between the future and present ( (CTRL2-SENS2B)-(CTRL1-SENS1B) ), not the differences of weather variables (in CTRL model) between the future and present (CTRL2-CTRL1). The current form of figure caption is a bit confusing.

---

## Author Comment (AC1) · 6 Dec 2019

Responses to referee #1:

General comments: In the manuscript 'Understanding Climate-Fire-Ecosystem Inter- actions Using CESM-RESFire and Implications for Decadal Climate Variability', Zou et al. explored complex interactions between climate change, fire, and ecosystem using a global Earth System Model equipped with a coupled fire module. They estimated the global net radiative effects and NEE changes due to fire aerosols and fire-induced land cover changes under present-day and future scenarios. The topic is interesting and relevant to the scope of ACP. Overall, this is a nicely written manuscript with a clear description of data, model design and results. I recommend it to be published after some minor modifications suggested below.

Response: Thank you for your recommendation and constructive comments. We revised the manuscript accordingly to improve the presentation quality. Please see below the point-by-point responses and corresponding revisions in the manuscript.

Specific comments: My only major concern is the present manuscript lacks a detailed discussion about the uncertainty of the simulations and calculations. Specifically, although most current state-of-art fire models (including RESFire used in this study) may be able to reproduce the main spatial variability of fire emissions (and fire pollutants) under current climate condition, their ability to simulate temporal variability, as well as the changes under a changing climate has not been validated. As mentioned by the authors, some important processes (such as the lightning changes in the warming future) are also ignored in this study. It will be interesting to know how does it lead to changes in the simulated fire impacts in the future scenario. I believe this paper will be benefited from adding some discussions on this topic.

Response: Thank you for the helpful suggestion. We agree with you that uncertainty is still a challenging issue for the current state-of-art fire models. The same statement also applies to global lightning projections under climate change scenarios. Before using the RESFire model for future projections, we comprehensively evaluated its modeling performance in terms of both spatial distributions and temporal variations for global burned area and fire emissions in our previously published model development paper in the Journal of Advances in Modeling Earth Systems (JAMES, Zou et al., 2019). As shown in the following figures (Figs. R1 and R2) reproduced from Figs. 9 and 10 of Zou et al. (2019), the RESFire model captures the burning patterns and fire seasonality in different regions driven by either reanalysis-based atmospheric data (RESFire_CRUNCEP) or online simulated atmospheric data (RESFire_CAM5). It can also reproduce the observed decadal trends driven by different forcing factors such as decadal climate variability as well as demographic and socioeconomic changes as shown in Andela and van der Werf (2014) and Andela et al. (2017). However, since climate-fire-ecosystem interactions are of interest in this work, we fixed the socioeconomic factors such as population density and GDP in the RESFire simulations to eliminate the uncertainties associated with future population and socioeconomic projections. Lightning was also fixed in the future projections due to large uncertainty in its parameterization and future projections (Tost et al., 2007; Clark et al., 2017). There are other considerable uncertainty factors remaining in the projections, including fire emission estimation, fire radiative forcing related with aerosol-cloud interactions, fire induced land cover change and biogeochemical/biophysical effects, etc. We added a new section 3.4 to discuss the relevant uncertainties you suggested.

[Figure]

*Figure R1 Comparisons of spatial distributions and seasonal variations of burned area in the observations and simulations. (a) GFED4.1s burned area fractions (%) averaged from 1997 to 2010; (b) seasonal variations of averaged GFED4.1s burned areas (km²) in the eight subregions; (c, d) same as (a, b) but from RESFire_CRUNCEPa driven by the CRUNCEP reanalysis-based atmospheric data and varying population density; (e, f) same as (a, b) but from RESFire_CRUNCEPb driven by the CRUNCEP reanalysis-based atmospheric data and fixed population density; (g, h) same as (a, b) but from RESFire_CAM5a driven by online bias corrected CAM5 atmosphere simulations and fixed population density; (i, j) same as (a, b) but from RESFire_CAM5b driven by online CAM5 atmosphere simulations without bias correction and fixed population density. The spatial correlation coefficients between simulated global burned area fractions and the GFED4.1s data are shown on the bottom left corners of (c), (e), (g), and (i). RESFire = REgion-Specific ecosystem feedback Fire; GFED = Global Fire Emissions Database; CRUNCEP = Climatic Research Unit and National Centers for Environmental Prediction; CAM5 = Community Atmosphere Model version 5. (reproduced from Fig. 9 in Zou et al., 2019)*

[Figure]

*Figure R2 Comparisons of decadal trends (%/year) in annual averaged burned areas from 1991 to 2010. (a) Burned area trends driven by natural and demographic forcing in RESFire_CRUNCEPa with changing weather and population; (b) burned area trends driven by only natural forcing in RESFire_CRUNCEPb with changing weather but fixed population density; (c) burned area trends driven by demographic changes only. RESFire = REgion- Specific ecosystem feedback Fire; CRUNCEP = Climatic Research Unit and National Centers for Environmental Prediction. (reproduced from Fig. 10 in Zou et al., 2019)*

Minor and technical comments:

Page 1, Line 17: "The complex climate-fire-ecosystem interactions were not included in previous climate model studies". I suggest softening the tune here. Some components of the interactions between climate, fire, and ecosystem have been considered in previous studies (although they were not necessarily incorporated into, or might not be represented thoroughly in a fully coupled online model).

Response: Thank you for the suggestion. We revised the narrative here to "The complex climate-fire-ecosystem interactions were not fully integrated in previous climate model studies".

Page 2, Line 58: "These processes were not included in previous climate model studies". Similar to the above, this sentence is way too assertive.

Response: Thank you. We revised it to "These processes were not fully included in previous climate model studies".

Page 3, Line 102-103: Since the new scheme is not implemented in this study (and the readers don't know the strength of the new approach), you don't have to mention it here. Removing this sentence won't affect the integrity of this paper.

Response: Thank you. The fire plume parameterization paper has been submitted to the Journal of Advances in Modeling Earth Systems and is under review now.

Page 7, Line 218-220: In addition to biogenic organic aerosols, can an underestimation of fire emissions be another reason for low simulated aerosols?

Response: You are right. We added this possible cause of underestimated fire emissions in line 231-232 of the revised manuscript as follows:

"Another possible cause for the underestimation problem is underrepresented burning activity due to deforestation and forest degradation and consequently underestimated fire aerosols emissions in these regions".

More detailed discussion is given in the next paragraph based on Fig. 2.

Page 7, Line 246-247: Any physical explanation for the differences between the signs of aerosol-cloud interactions and aerosol-radiation interactions?

Response: As explained in line 252-256, the land-sea contrast warming and cooling effects by aerosol-radiation interactions over Africa and South America (Fig. 3a) result from strong light absorption of fire aerosols enhanced by increased low-level cloud reflection over the downwind ocean areas. Fig. R3 shows the changes in low-level cloud fractions induced by fire aerosols in the present-day simulation (CTRL1-SENS1A). It demonstrates decreased low-level clouds over the African land region where biomass burning occurs and increased low-level clouds over the downwind Atlantic Ocean region. Therefore, opposite land-sea contrast signs occur due to distinct aerosol-cloud and aerosol-radiation interactions with positive aerosol-cloud radiative forcing over the land region and negative aerosol-cloud radiative forcing over the ocean area (Fig. 3b).

We added Fig. R3 in the supplement and more detailed explanation in line 259-265:
"The land-sea contrast of radiative effects emerges again in the vicinity of Africa and South America, but the signs of the contrasting effect related with aerosol-cloud interactions are opposite to these from aerosol-radiation interactions. The large amounts of fire aerosols suppress

low-level clouds over the African land region by stabilizing the lower atmosphere through reduction of radiative heating of the surface. However, fire aerosols increase cloud cover and brightness in the downwind Atlantic Ocean areas because they increase the number of cloud condensation nuclei and the larger cloud droplet number density reduce cloud droplet sizes (Lu et al., 2018; Rosenfeld et al., 2019; Fig. S1 in the Supplement)".

[Figure]

*Figure R3 Fire aerosol induced low-level cloud fraction change (unit: %) in the CESM-RESFire present-day simulation (CTRL1-SENS1A).*

Page 8, Line 279: It would be good to briefly introduce this plume rise parameterization (e.g., based on what measurements? Global universal or regional-based?)

Response: Thank you for the suggestion. The Sofiev et al. (2012) plume rise parameterization is globally universal and is based on atmospheric boundary layer height, fire radiative power, and Brunt-Väisälä frequency in the free troposphere. We added relevant description in line 299-302 as follows:

"In our simulations, we used a simplified plume rise parameterization (Sofiev et al., 2012) based on online calculated fire burning intensity (FRP) and atmospheric stability conditions (PBLH and Brunt-Väisälä frequency) in CESM-RESFire and applied vertical profiles with diurnal cycles to the vertical distribution of fire emissions".

Page 11, Line 376-379: The terms 'fire combustion factors', 'fire spread distribution', and 'fire spread factors' are probably not familiar to many readers. Please consider a short explanation on these parameters (i.e., what do they mean physically).

Response: Thanks. Fire combustion factors (FCF) are based on 10-day running mean of surface temperature, 10-day running mean of total precipitation, and soil water fraction for top 0.05 m layers as a surrogate for fuel combustibility (see Table 3 of Zou et al. (2019)), while fire spread factors (FSF) include surface air temperature, relative humidity, surface soil wetness, and fractions of wet canopy as listed in Table 4 of Zou et al. (2019). We added the explanation for these terms in line 396-397 and line 400-401 of the revised manuscript.

Page 11, Line 388-389: I don't quite understand the causal relationship stated in this sentence. The changes in wind speed are higher over the ocean than that over land, but this could be

simply due to the larger magnitude of wind speed over the ocean. Relatively smaller changes in land wind speed could still have large impacts on fire spread and burned area.
Response: Thank you for the comment. We rewrote the analysis for climate-fire-ecosystem interactions in Sect. 3.3. Please see the revised manuscript for details.

Page 25, Figure 2: Please align tick label '0.1' with other tick labels in panels b, c, d.
Response: Thanks. The figure has been updated.

Page 27, Line 817: Should the unit of CDNUMC '10^9 # /m2' (as correctly shown in panel d)?
Response: That's correct. Thanks for the correction.

Page 30, Figure 7: The colors in panel c don't have enough separation. Please use another scale.
Response: Thank you. The scale in this figure has been updated for better color separation.

Page 32, Figure 9: If my understanding is correct, the data in this figure show the differences of fire modifications on weather variables between the future and present ( (CTRL2-SENS2B)-(CTRL1-SENS1B) ), not the differences of weather variables (in CTRL model) between the future and present (CTRL2-CTRL1). The current form of figure caption is a bit confusing.
Response: Thank you for the correction. Fig. 9 is used to explain the future changes in simulated global fire activity. Therefore, we compare the future and present-day fire weather conditions in CTRL2 and CTRL1 to understand these fire simulation results shown in Figs. 7 and 8. This figure has been updated with the corresponding sensitivities to surface temperature, precipitation, relative humidity, and surface wind speed between CTRL2 and CTRL1. The changes in fire feedback on these fire weather variables (i.e., (CTRL2-SENS2B)-(CTRL1-SENS1B)) are shown in Fig. 10 with corresponding discussion in Sect. 3.3 of the revised main text.

References

Andela, N., Morton, D. C., Giglio, L., Chen, Y., van der Werf, G. R., Kasibhatla, P. S., DeFries, R. S., Collatz, G. J., Hantson, S., Kloster, S., Bachelet, D., Forrest, M., Lasslop, G., Li, F., Mangeon, S., Melton, J. R., Yue, C., Randerson, J. T.: A human-driven decline in global burned area, Science, 356(6345), 1356–1362. https://doi.org/10.1126/science.aal4108, 2017.

Andela, N., and van der Werf, G. R.: Recent trends in African fires driven by cropland expansion and El Nino to La Nina transition, Nature Climate Change, 4, 791–795. https://doi.org/10.1038/nclimate2313, 2014.

Clark, S. K., Ward, D. S., and Mahowald, N. M.: Parameterization-based uncertainty in future lightning flash density, Geophys. Res. Lett., 44, 2893–2901, doi:10.1002/2017GL073017, 2017.

Sofiev, M., Ermakova, T., and Vankevich, R.: Evaluation of the smoke-injection height from wild-land fires using remote-sensing data, Atmos. Chem. Phys., 12, 1995-2006, 10.5194/acp-12-1995-2012, 2012.

Tost, H., Jöckel, P., and Lelieveld, J.: Lightning and convection parameterisations – uncertainties in global modelling, Atmos. Chem. Phys., 7, 4553–4568, https://doi.org/10.5194/acp-7-4553-2007, 2007.

Zou, Y., Wang, Y., Ke, Z., Tian, H., Yang, J., and Liu, Y.: Development of a REgion-Specific ecosystem feedback Fire (RESFire) model in the Community Earth System Model, J. Adv. Model Earth Sy., https://doi.org/10.1029/2018MS001368, 2019.

---

## Author Comment (AC2) · 6 Dec 2019

Responses to referee #2:

The manuscript by Zou et al. presents an analysis of the interactions between climate, wildfires, ecosystems, and radiative balance in a recently (further) developed modelling system, CESM-RESFire. The methodology includes a suitable set of sensitivity experiments that provide substantial new insight into the role of different types of potential interactions (mainly aerosol effects and land cover changes) in driving present-day radiative effects of wildfires, and their future radiative forcing. It features some novel aspects compared to previous studies, especially when it comes to the types of feedbacks allowed and investigated, and provides a useful contribution to the improvement of our poor understanding of the role of fire in the Earth system. The manuscript is nicely written, and well within the scope of Atmospheric Chemistry and Physics. I find it worthy of publication, following some (mostly minor) improvements that I describe below.

Response: Thank you for your constructive comments and helpful suggestions. We revised the manuscript following the suggested improvements. Please see below the point-by-point responses with corresponding revisions in the manuscript.

GENERAL COMMENTS:
- The title, abstract and conclusions (as well as the main text) leave the reader thinking that the full climate effects of wildfires are examined in the current study. However, this is somewhat misleading, as full climate responses (i.e. temperature, precipitation, humidity etc changes) are not explored or discussed, even if they are partially included (I am saying "partially" since the oceans and sea-ice are fixed). The study goes up to radiative effect and radiative forcing quantification, and that should be reflected more accurately in the different parts of the text. In my specific comments below, there are some suggestions for amending this, but the authors should make an effort to do so further throughout the text.
Response: Thank you for the suggestion. We added two new figures (Fig. 9 and 10) and relevant discussion in Sect. 3.3 of the main text to examine future changes in full climate responses (Fig.9: CTRL2-CTRL1) of fire weather variables (surface air temperature, precipitation, relative humidity, and surface wind speed) as well as these associated with fire feedbacks (Fig. 10: (CTRL2-SENS2B)-(CTRL1-SENS1B)) to explore fire-climate interactions. The fire weather changes without fire feedbacks (SENS2B-SENS1B) are shown in Fig. S3 in the supplement for comparison. Please see our responses to your specific comments below for more details.

- In addition, the future radiative impacts (whose study presumably is a core aim of this work) are discussed very briefly towards the end of Sect. 3.3, and in a way that does not seem accurate/consistent with what is shown on the maps (see related comment below).
Response: Thank you. We rewrote the major part in Sect. 3.3 with more detailed and consistent discussion for climate-fire-ecosystem interactions and fire radiative forcing. Please see Sect. 3.3 in the revised manuscript for details.

- The past tense is often used in the text to refer to the work presented here, where the present tense would be more appropriate/standard. For example "We provided a brief model description and sensitivity experiment settings in Section 2. . .", where "provide" would probably read better. I suggest making this amendment to wherever applicable in the text.

Response: Thank you. We made the amendments with proper tense throughout the manuscript as suggested for better presentation.

SPECIFIC COMMENTS:
Page 2, Lines 1-3: I suggest changing the title to "Using CESM-RESFire to Understand Climate-Fire-Ecosystem Interactions and their Implications for Radiative Forcing". The title as it stands currently is misleading, as "implications for decadal climate variability" were not examined at all in this study. Generally speaking, it is radiative forcing/effects that were examined, rather than climate (temperature, humidity, precipitation etc) effects.

Response: Thank you for the suggestion. In our response to the above general comments, we added two new figures (Fig. 9 and 10) in the main text and one more figure (Fig. S3) in the supplement to fully evaluate fire weather and climate responses to different driving factors such as increased GHG concentrations and fire feedback through multiple pathways. Besides, we also discussed the carbon budget response to fire disturbances in the present-day and future scenarios in Sect. 3.2, which is one of the major research objectives for the comprehensive evaluation of climate-fire-ecosystem interactions and future projections in this work. Radiative forcing and carbon budget were used as two evaluation metrics for these objectives. Therefore, we partially accepted your suggestion and changed the title to "Using CESM-RESFire to Understand Climate-Fire-Ecosystem Interactions and the Implications for Decadal Climate Variability".

Page 1, Line 20: For the same reason, I suggest rephrasing to ". . .and their impacts on fire activity and radiative forcing".

Response: Thank you for the suggestion. We discussed fire impacts on carbon budget in Sect. 3.2. We also added more figures and discussion in Sect. 3.3 of the revised manuscript to improve our analysis on fire feedbacks to the climate systems and fire activity itself.

Page 2, Line 38: Please add ", respectively" at the end of the sentence.

Response: Thank you. It's added.

Page 2, Lines 57-58: "are further confounded by natural processes and human interferences" – human and natural processes have been mentioned in the previous sentence. Why repeat them?

Response: Thank you. We removed this sentence as suggested.

Page 2, Line 69: "used the same approach" – suggest changing to "used the same unidirectional approach".

Response: Thank you. We added "unidirectional" in this sentence as suggested.

Page 2, Line 71: The term "fixed" may not be fully accurate here. For example, to my knowledge, Tosca et al. (2013) performed simulations with and without aerosol emissions, with no "fixing" per se involved.

Response: Thank you. Tosca et al. (2013) used monthly cycling 1997–2009 fire emissions based on the GFEDv3 dataset. Therefore, we changed "fixed" here to "prescribed".

Page 3, Line 79: I do not think "feedback in" is needed.

Response: Thank you. It's removed.

Page 3, Lines 81-84: Change past tense to present tense (just another example).
Response: Thank you. We changed past tense to present tense wherever applicable in the manuscript.

Sect. 2.1: Gas-phase chemistry (e.g. ozone and its precursors) is not mentioned at all in the model description – or anywhere really. If such a mechanism is not included, this should be mentioned (along with acknowledging the potentially sizeable effect of this missing process), and if included, the authors should describe in what fashion it is included.
Response: Thank you. We used a no gas-phase chemistry version of the CAM5 model with prescribed $O_3$ concentrations. Therefore, we didn't consider photochemical reactions and chemistry-climate feedbacks in our discussion. We added an explanation in line 102-103 as follows: "The gas-phase photochemistry is not included in the CAM5 simulations, which precludes the possibility for evaluating chemistry-climate interactions".

Sect. 2.1: No mention at all of biogenic aerosols.
Response: In CAM5, biogenic emissions of volatile organic compounds (isoprene, monoterpenes, toluene, big alkenes, and big alkanes) are pre-defined and read from an input emission file derived from MOZART VOC emissions. Then a simple treatment of secondary organic aerosol (SOA) is used to assume fixed mass yields (Table R1) for anthropogenic and biogenic precursor VOCs (Neale et al., 2013). The total yielded mass is emitted as the SOA (gas) species and condensation/evaporation of the SOA (gas) to/from three aerosol modes are calculated in the MAM3 module.

We added the description of biogenic aerosols in CAM5 in line 98-102 of this section: "A simple treatment of secondary organic aerosols (SOA) is used in CAM5 to derive SOA formation from anthropogenic and biogenic volatile organic compounds (VOCs) with fixed mass fields (Table S1 in the Supplement). The total SOA mass is emitted as the SOA (gas) species from the surface and then condensation/evaporation of gas-phase SOA to/from different aerosol modes are calculated in the MAM3 module (Neale et al., 2013)", and the information of input files in line 142-144 of Sect. 2.2: "Emission fluxes for the 5 VOC species (isoprene, monoterpenes, toluene, big alkenes, and big alkanes) to derive SOA mass yields were prescribed from the MOZART-2 dataset (Horowitz et al., 2003)".
We also added a new table (Table S1) of SOA (gas) mass yields in the supplement.

*Table R1 Assumed SOA (gas) yield in CAM5*

| Species | Mass yield | Reference |
|---|---|---|
| Big Alkanes | 5% | Lim and Ziemann (2005) |
| Big Alkenes | 5% | Assumed |
| Toluene | 15% | Odum et al. (1997) |
| Isoprene | 4% | Kroll et al. (2006) |
| Monoterpenes | 25% | Ng et al. (2007) |

Page 3, Line 97: Probably "microphysics" and not "macrophysics"?

Response: The parameterization of stratiform cloud microphysics is based on Morrison and Gettlman (2008). Its implementation in the CAM model is described in Gettelman et al. (2008). Cloud macrophysics is suite of physical processes that computes (1) cloud fractions in each layer; (2) horizontal and vertical overlapping structures of clouds; (3) net conversion rates of water vapor into cloud condensates. Its parameterization and implementation in CAM5 are described in Park et al., 2014.

We added the cloud microphysics scheme in line 97-98 of the revised manuscript. Please refer to Sect. 4.6 "Cloud Microphysics" and Sect. 4.7 "Cloud Macrophysics" of the CAM5 Tech Notes (Neale et al., 2013) for more details.

Page 4, Line 110: Please mention the year for the future scenario. It's mentioned later, but worth mentioning it here too.
Response: Thank you. We added the modeling years for both scenarios here.

Page 4, Lines 138-140: Suggested rephrasing – ". . .we allowed the semi-static historical LCC data for the year 2000 from the version 1 of the Land-Use History A product (LUHa.v1) (Hurtt et al., 2006) to be affected by post-fire vegetation changes (Zou et al., 2019)".
Response: Thank you. We revised the sentence as suggested.

Page 5, Lines 150-151: "given great uncertainties in future projections of these inputs" - Are these uncertainties larger than for the rest of the variables considered here?
Response: Lightning is critical for atmospheric chemistry and wildfire simulations. Its parameterizations in global models differ greatly among the models and have very large uncertainties. Several studies have compared different lightning parameterizations and evaluated the uncertainties in future projections. For instance, Tost et al. (2007) compared different combinations of convection and lightning parameterizations with satellite observations and found a wide range in the spatial and temporal variability of the simulated lightning flash density. Similarly, Clark et al. (2017) evaluated the performance of 8 lightning parameterizations in CAM5 and tested the sensitivity of future lightning activity to the choice of parameterization. They found that future changes in global mean lightning flash density are highly sensitive to the parameterization chosen, with cloud top height schemes, a cold cloud depth scheme, and a scheme based on convective mass flux projecting large increases (36% to 45%), a mild increase (12.6%), and a decrease (−6.7%) in lightning flash density, respectively, under the RCP8.5 scenario. Finney et al. (2018) got a similar conclusion by comparing a new upward cloud ice flux (IFLUX) approach with the widely used cloud-top height (CTH) approach, with a 15% decrease in total lightning flash rate in 2100 under RCP8.5 based on IFLUX in contrast to previously reported global increase in lightning based on CTH. They also identified the largest differences in the tropics where most lightning and fires occur.

Moreover, these regions are also heavily affected by anthropogenic activities such as deforestation and agriculture expansion (Andela et al., 2017;Morton et al., 2008;Van der Werf et al., 2010). The future changes in human activities and associated land use and land cover change are also strongly dependent on the choice of different socioeconomic development pathways (Riahi et al., 2017).

In this work, we decided to use observation-based lightning and present-day demographic data in our fire simulations in order to focus on climate-fire-ecosystem interactions that are of interest of this study. We revised the discussion in line 159-164 to "It is worth noting that we used the present-day demographic data and observation-based climatological lightning data in the future scenario given pathway-dependence and great uncertainties in future projections of these inputs (Clark et al., 2017;Riahi et al., 2017;Tost et al., 2007;). In other words, we did not consider the influence of fire ignition changes associated with human activity or lightning flash density in our future projection simulations but focused on broad impacts of future climate change on fuel loads and combustibility as well as fire weather conditions", and added a new section 3.4 to discuss the relevant uncertainties.

Page 5, Lines 161-164: Yes, but are the timescales long enough in this case for this assumption to hold? Please discuss.

Response: In our fire model development study (Zou et al., 2019), we evaluated the post-fire temporal evolution of carbon budget variables in different regions. Here we reproduced Fig. S3 in Zou et al. (2019) to show these changes in an idealized burning experiment, in which we conducted single-year burning events in fire peak months of 7 typical fire-prone regions. The post-fire recovery rates in net ecosystem productivity (NEP, g C m$^{-2}$ yr$^{-1}$, positive for net ecosystem carbon uptake) vary among different PFT groups with the mean recovery periods of post-fire NEP about 3–18 years. For most regions, the recovery time is less than or about the simulation time period in this study. Therefore, we consider this assumption valid in general. Since ocean and terrestrial carbon sinks are not simulated in this study, we had to rely on a "concentration-driven" approach instead of the "emission-driven" one by prescribing atmospheric $CO_2$ concentrations in each modeling scenario. A series of comprehensive assessments of the global carbon cycle are available by using observation-constrained modeling estimates of all carbon sources and sinks (Le Quéré et al., 2018a; Le Quéré et al., 2018b), which is out of the scope of this study.

[Figure]

*Figure R1. Simulated post-fire temporal evolution of carbon budget in different PFT regions based on an idealized burning experiment. (a) spatial distributions of annual averaged NEP (gC m⁻² yr⁻¹); (b) temporal variations of post-fire NEP in each disturbed mode grid cell. tmp_shrub: temperate shrub dominated; trp_tree: tropical forest dominated; brl_tree: boreal forest dominated; tmp_tree: temperate forest dominated. (reproduced from Fig. S3 in Zou et al., 2019)*

Page 5, Line 175: "the Ghan's method" -> "the Ghan method"
Response: Thank you. It is changed as suggested.

Equations (1): The way these equations are written is very confusing. First of all because of the dashes ("-") and the minuses appearing identical, and also because of the use of column (:). I suggest the following format:
"RE of interaction of radiation with fire aerosol: RE = Δ(F -F)" (with the appropriate subscripts in each case)
Response: Thank you. We revised the equations as suggested.

Page 6, Line 188: "nonnegligible" -> "non-negligible"
Response: Corrected. Thanks.

Page 6, Line 200: "budge" -> "budget"
Response: Corrected. Thanks.

Page 7, Lines 220-221: "However, the model well captured the high AOD regions over the Northern and Southern Hemispheres of Africa" – I am not sure I see this on Fig. 1. Therefore the statement seems too confidently positive.
Response: Thank you for the comment. We revised the description in line 232-235 to "The AOD simulations over tropical savanna regions with pervasive biomass burning activities are also lower than the satellite observations, which might be attributable to both underestimated online fire emissions and too strong wet scavenging of primary carbonaceous aerosols in the CAM5-MAM3 model (Liu et al., 2012)".

Page 7, Line 228: The AERONET measurements cannot be characterised as "in situ". They are also remotely sensed.
Response: Corrected. Thanks.

Fig. 3: Please specify that this is TOA radiative effect.
Response: The description was added in the figure caption. Thanks.

Page 7, Lines 243-247: There are some areas that experience pronounced positive forcing due to fire aerosol-cloud interactions. The most prominent ones are Europe and most of Africa. Presumably that is because of black carbon stabilization effects? But why would these be more important in these specific regions? Any thoughts? Please comment.
Response: Fig. R2 shows the changes in low-level cloud fractions induced by fire aerosols in the present-day simulation (CTRL1-SENS1A). It demonstrates decreased low-level clouds over Europe and most of African land regions and increased clouds over most of the other regions. Therefore, positive radiative forcing is found over these land regions concurrent with reduced cloud coverage in contrast to other regions with increased cloud coverage (Fig. 3b in the manuscript). Though these decreased clouds are not statistically significant, the cooling of the surface due to fire aerosol scattering stabilizes the lower atmosphere and reduces cloud formation. We added Fig. S1 in the supplement and added more detailed explanation in line 261-265 as follows:
"The large amounts of fire aerosols suppress low-level clouds over the African land region by stabilizing the lower atmosphere through reduction of radiative heating of the surface. However, fire aerosols increase cloud cover and brightness in the downwind Atlantic Ocean areas because they increase the number of cloud condensation nuclei and the larger cloud droplet number density reduce cloud droplet sizes (Lu et al., 2018; Rosenfeld et al., 2019; Fig. S1 in the Supplement)".

[Figure]

*Figure R2 Fire aerosol induced low-level cloud fraction change (unit: %) in the CESM-RESFire present-day simulation (CTRL1-SENS1A). The net meshes denote the 0.05 significance level.*

Page 8, Lines 248-249: Why are there areas with both positive and negative changes? Why is Africa pretty much all negative? These are interesting features. Please elaborate.

Response: The estimate of the radiative effect associated with fire aerosol-induced surface albedo change is based on the Ghan method (2013), which considers both changes in snow albedo due to deposition of light-absorbing fire aerosol, and changes in snow cover induced by fire aerosol caused precipitation change. As shown by Fig. R3a, the snow depths decrease in most Arctic regions in CTRL1, suggesting snow albedo reduction due to deposition of absorptive aerosols to snow. In contrast, these regions in Canada, eastern Siberia, and Tibet show increased surface albedo in Fig. R3b, which results from increases in snowfall and snow cover over these regions. These surface albedo changes modulate the reflection of incoming solar radiation and finally alter the net shortwave radiative flux at the TOA (Fig. 3c in the manuscript). The fire aerosol-induced albedo effect is more significant in high-latitude regions than others because of the spatial distribution of snow cover and snow precipitation. Other factors like numerical noise might also contribute to these simulated radiation changes between the two experiments. We added the above discussion in line 265-270 and Fig. S2 in the supplement.

[Figure]

*Figure R3 Fire aerosol induced snow and albedo changes between CTRL1 and SENS1A. (a) changes in snow depths (m) over ice; (b) changes in surface albedo (unitless). The net meshes denote the 0.05 significance level.*

Page 8, Lines 262-263: Please specify that the Jiang et al. (2016) study was performed with the same atmospheric model as in the current study (though older version?), as it is useful for the reader to know.

Response: Thank you. Jiang et al. (2016) used the same version of the CAM5 model (CAM5 version 5.3) but with a 4-mode modal aerosol module (MAM4). We added this information in line 282-284.

Page 9, Line
 293: I am not sure where the +51% value comes from. From Table 2, the Raci is -1.31 for 2050 in this study and -1.42 in the CCSM study. Or do the authors mean something different and I am missing the point? In any case, I think it should be made clearer where the +51% value comes from.

Response: Here the percentage indicates the change in net fire radiative forcing estimated by Ward et al. (2012), which increases by 51% from -0.55 W/m$^2$ in the 2000s to -0.83 W/m$^2$ in the 2100s based on the CCSM forcing data. We revised the discussion in line 471-473 to "This projection result is larger than the change in net fire radiative forcing based on the CCSM future projection in Ward et al. (2012), which suggested a 51% increase from -0.55 W m$^{-2}$ in the 2000s to -0.83 W m$^{-2}$ in the 2100s (Table 2)".

Page 9, Lines 313-315: "Such difference is also consistent with the changes in different versions of the GFED datasets, which show a 11% increase of global fire carbon emissions in the latest GFED4s as compared with the old GFED3 for the overlapping 1997-2011 time period (van der Werf et al., 2017)" – Do the authors mean that there is an upward "trend" between older and newer GFED emissions versions, implying that eventually the GFED emissions will match the online model? That's a rather simplistic reasoning and needs to be supported further or phrased differently.

Response: The difference of fire carbon emissions between the new GFED4s data and the old GFED3 data is a result of estimation changes in both global burned area and mean fuel consumption (van der Werf et al., 2017). The GFED4s data includes burned area and emission estimates from small fires that are missing in the old GFED3 data. These small fires were difficult to be resolved by satellite remote sensing techniques before. The new dataset used a revised version of the Randerson et al. (2012) small-fire estimation approach to include these small fires. We added the explanation in line 331-333 as "This increased global fire carbon emissions in the GFED4s dataset result from a substantial increase in global burned area (+37%) due to inclusion of small fires and a modest decrease in mean fuel consumption (-19%) according to van der Werf et al. (2017)".

Page 11, Lines 361-363: "Though we mainly focused on fire-climate interactions without consideration of human impacts in this study, the RESFire model is capable of reproducing the anthropogenic interference on fire activity as observed from the space (Zou et al., 2019)" – This needs some more explanation. The common understanding is that in Northern Hemisphere Africa the decline in burned area is due to agricultural conversion and resulting landscape fragmentation (e.g. Andela et al., 2017). Is this a process that is represented in this particular model? Please clarify and discuss.

Response: Before using the RESFire model for future projections, we comprehensively evaluated its modeling performance in terms of both spatial distributions and temporal variations for global burned area and fire emissions in our previously published model development paper (Zou et al., 2019). The RESFire model is capable to reproduce the observed decadal trends driven by different forcing factors such as decadal climate variability as well as demographic and socioeconomic changes (as shown in Fig. R3 and in Andela and van der Werf, 2014 and Andela et al., 2017). However, since climate-fire-ecosystem interactions are of interest in this work, we fixed socioeconomic factors such as population density and GDP in the RESFire simulations and projections. We added more explanation and discussion in line 379-382 as follows:

"Though we mainly focus on fire-climate interactions without consideration of human impacts in this study, the RESFire model is capable of capturing the anthropogenic interference on fire activity and reproducing observation-based long-term trends of regional burning activity driven by climate change and human factors (Zou et al., 2019)".

[Figure]

*Figure R4 Comparisons of decadal trends (%/year) in annual averaged burned areas from 1991 to 2010. (a) Burned area trends driven by natural and demographic forcing in RESFire_CRUNCEPa with changing weather and population; (b) burned area trends driven by only natural forcing in RESFire_CRUNCEPb with changing weather but fixed population density; (c) burned area trends driven by demographic changes only. RESFire = REgion- Specific ecosystem feedback Fire; CRUNCEP = Climatic Research Unit and National Centers for Environmental Prediction. (reproduced from Fig. 10 in Zou et al., 2019)*

Page 11, Lines 390-394: The evidence to support this statement is somewhat weak. First of all, the precipitation changes (Fig. 9c) are not significant almost everywhere (therefore, not much difference in that respect to the wind changes). Secondly, the match between locations with decreased precipitation and increased burned area (and the other way around) is not always clear (e.g. the north of Siberia experiences increases in burned area, but simultaneous increases in precipitation; there are other examples too). It would be best to discuss this in a more quantitative fashion, e.g. report the spatial correlation coefficients between burned area and driver variables to extract more robust conclusions?

Response: Thank you for the suggestion. We added the spatial correlation coefficients in line 403-404. We also rewrote the major part of Sect. 3.3 with more detailed discussion of climate-fire-ecosystem interactions. Please see the revised manuscript for details.

Figure 11: This figure is not really discussed in any insightful way, beyond just stating that such effects "might compensate biogeochemical warming effects of deforestation related carbon-cycle changes". How does each individual variable shown affect warming/cooling patterns, and which of these variables seems to be more important, based on this analysis?

Response: The fire related albedo change and radiative effect are not as significant as suggested by previous studies. Therefore, we moved this figure from the main text to the supplement (Fig. S6) in the revised manuscript and added corresponding discussion in line 461-465 as follows: "Previous studies have suggested a net cooling effect of deforestation that could compensate for GHG waring effects on a global scale (Bala et al., 2007;Jin et al., 2012;Randerson et al., 2006). Though our model captures the reduction of forest coverage and increased springtime albedo in high-latitude regions (Fig. S6 in the Supplement), the radiative effect of fire induced LCC is almost neutral on a global basis in both present-day and future scenarios (Table 2)".

Page 12, Lines 427-431: This discussion is rushed and I am not sure I follow the reasoning. It is stated that the radiative forcing of aerosol-radiation interactions "show similar patterns with Fig. 3a, with generally cooling effects over the vicinities of fire areas and warming effects over the downwind regions". Where do we see this? In Fig. 3a, this was evident e.g. in and around Africa (and possibly South America), but I cannot see this in Fig. 12c. Then for aerosol-cloud interactions, it is stated that there are "warming effects in Southeast Asia and Australia due to local cloud changes", but how are these features consistent with Fig. 3b, in which the inclusion of fire caused negative radiative effects due to aerosol-cloud interactions over those regions (as was the case for northern high latitudes).

Response: We rewrote the major part of Sect. 3.3 with more detailed and consistent discussion of climate-fire-ecosystem interactions. Please see the revised manuscript for details.

Page 13, Line 447: Please add "fire" between "significant" and "aerosol".

Response: The cooling effect is mainly contributed by aerosols from anthropogenic and industrial emission sources in the eastern U.S. and China. Therefore, we prefer to keep it as is to be consistent with the references cited here.

Page 13, Lines 455-456: Please change "climate effects" to "radiative effects", as the former implies that effects on temperature, precipitation etc due to fires were also examined (which is not the case).

Response: We discussed fire feedback effects on climate and weather variables such as air temperature, precipitation, relative humidity, and surface wind speed in Fig. 10 and line 419-428 of the revised manuscript. Please see Sect. 3.3 for more details.

Page 13, Line 465: Please change "their" to "its".

Response: Thanks. This sentence is changed to "More evaluation metrics such as large wildfire extreme events should be considered in future studies to improve our understanding of global and regional fire activities, their variations and trends, and their relationship with decadal climate change.".

References

Andela, N., and van der Werf, G. R.: Recent trends in African fires driven by cropland expansion and El Nino to La Nina transition, Nature Climate Change, 4, 791–795. https://doi.org/10.1038/nclimate2313, 2014.

Andela, N., Morton, D.C., Giglio, L., Chen, Y., Van Der Werf, G.R., Kasibhatla, P.S., DeFries, R.S., Collatz, G.J., Hantson, S., Kloster, S. and Bachelet, D.: A human-driven decline in global burned area. Science, 356(6345), 1356-1362, 2017.

Clark, S.K., Ward, D.S. and Mahowald, N.M.: Parameterization-based uncertainty in future lightning flash density. Geophysical Research Letters, 44(6), 2893-2901, 2017.

Finney, D.L., Doherty, R.M., Wild, O., Stevenson, D.S., MacKenzie, I.A. and Blyth, A.M.: A projected decrease in lightning under climate change. Nature Climate Change, 8(3), 210, 2018.

Ghan, S. J.: Technical Note: Estimating aerosol effects on cloud radiative forcing, Atmos. Chem. Phys., 13, 9971–9974, https://doi.org/10.5194/acp-13-9971-2013, 2013.

Gettelman, A., Morrison, H. and Ghan, S.J.: A new two-moment bulk stratiform cloud microphysics scheme in the Community Atmosphere Model, version 3 (CAM3). Part II: Single-column and global results. Journal of Climate, 21(15), 3660-3679, 2008.

Kroll, J.H., Ng, N.L., Murphy, S.M., Flagan, R.C. and Seinfeld, J.H.: Secondary organic aerosol formation from isoprene photooxidation. Environmental science & technology, 40(6), 1869-1877, 2006.

Le Quéré, C., Andrew, R. M., Friedlingstein, P., Sitch, S., Pongratz, J., Manning, A. C., Korsbakken, J. I., Peters, G. P., Canadell, J. G., Jackson, R. B., Boden, T. A., Tans, P. P., Andrews, O. D., Arora, V. K., Bakker, D. C. E., Barbero, L., Becker, M., Betts, R. A., Bopp, L., Chevallier, F., Chini, L. P., Ciais, P., Cosca, C. E., Cross, J., Currie, K., Gasser, T., Harris, I., Hauck, J., Haverd, V., Houghton, R. A., Hunt, C. W., Hurtt, G., Ilyina, T., Jain, A. K., Kato, E., Kautz, M., Keeling, R. F., Klein Goldewijk, K., Körtzinger, A., Landschützer, P., Lefèvre, N., Lenton, A., Lienert, S., Lima, I., Lombardozzi, D., Metzl, N., Millero, F., Monteiro, P. M. S., Munro, D. R., Nabel, J. E. M. S., Nakaoka, S., Nojiri, Y., Padin, X. A.,

Peregon, A., Pfeil, B., Pierrot, D., Poulter, B., Rehder, G., Reimer, J., Rödenbeck, C., Schwinger, J., Séférian, R., Skjelvan, I., Stocker, B. D., Tian, H., Tilbrook, B., Tubiello, F. N., van der Laan-Luijkx, I. T., van der Werf, G. R., van Heuven, S., Viovy, N., Vuichard, N., Walker, A. P., Watson, A. J., Wiltshire, A. J., Zaehle, S., and Zhu, D.: Global Carbon Budget 2017, Earth Syst. Sci. Data, 10, 405–448, https://doi.org/10.5194/essd-10-405-2018, 2018a.

Le Quéré, C., Andrew, R. M., Friedlingstein, P., Sitch, S., Hauck, J., Pongratz, J., Pickers, P. A., Korsbakken, J. I., Peters, G. P., Canadell, J. G., Arneth, A., Arora, V. K., Barbero, L., Bastos, A., Bopp, L., Chevallier, F., Chini, L. P., Ciais, P., Doney, S. C., Gkritzalis, T., Goll, D. S., Harris, I., Haverd, V., Hoffman, F. M., Hoppema, M., Houghton, R. A., Hurtt, G., Ilyina, T., Jain, A. K., Johannessen, T., Jones, C. D., Kato, E., Keeling, R. F., Goldewijk, K. K., Landschützer, P., Lefèvre, N., Lienert, S., Liu, Z., Lombardozzi, D., Metzl, N., Munro, D. R., Nabel, J. E. M. S., Nakaoka, S., Neill, C., Olsen, A., Ono, T., Patra, P., Peregon, A., Peters, W., Peylin, P., Pfeil, B., Pierrot, D., Poulter, B., Rehder, G., Resplandy, L., Robertson, E., Rocher, M., Rödenbeck, C., Schuster, U., Schwinger, J., Séférian, R., Skjelvan, I., Steinhoff, T., Sutton, A., Tans, P. P., Tian, H., Tilbrook, B., Tubiello, F. N., van der Laan-Luijkx, I. T., van der Werf, G. R., Viovy, N., Walker, A. P., Wiltshire, A. J., Wright, R., Zaehle, S., and Zheng, B.: Global Carbon Budget 2018, Earth Syst. Sci. Data, 10, 2141–2194, https://doi.org/10.5194/essd-10-2141-2018, 2018b.

Lim, Y.B. and Ziemann, P.J.: Products and mechanism of secondary organic aerosol formation from reactions of n-alkanes with OH radicals in the presence of NOx. Environmental science & technology, 39(23), 9229-9236, 2005.

Morton, D.C., Defries, R.S., Randerson, J.T., Giglio, L., Schroeder, W. and Van Der Werf, G.R.: Agricultural intensification increases deforestation fire activity in Amazonia. Global Change Biology, 14(10), 2262-2275, 2008.

Morrison, H. and Gettelman, A.: A new two-moment bulk stratiform cloud microphysics scheme in the Community Atmosphere Model, version 3 (CAM3). Part I: Description and numerical tests. Journal of Climate, 21(15), 3642-3659, 2008.

Neale, R. B., Chen, C. C., Gettelman, A., Lauritzen, P. H., Park, S., Williamson, D. L., Conley, A. J., Garcia, R., Kinnison, D., Lamarque, J. F., Marsh, D., Mills, M., Smith, A. K., Tilmes, S., Vitt, F., Morrison, H., Cameron-Smith, P., Collins, W. D., Iacono, M. J., Easter, R. C., Ghan, S. J., Liu, X. H., Rasch, P. J., and Taylor, M. A.: Description of the NCAR Community Atmosphere Model (CAM 5.0), NCAR 289, 2013.

Ng, N.L., Chhabra, P.S., Chan, A.W.H., Surratt, J.D., Kroll, J.H., Kwan, A.J., McCabe, D.C., Wennberg, P.O., Sorooshian, A., Murphy, S.M. and Dalleska, N.F.: Effect of NOx level on secondary organic aerosol (SOA) formation from the photooxidation of terpenes. Atmospheric Chemistry and Physics, 7(19), 5159-5174, 2007.

Odum, J.R., Jungkamp, T.P.W., Griffin, R.J., Forstner, H.J.L., Flagan, R.C. and Seinfeld, J.H.: Aromatics, reformulated gasoline, and atmospheric organic aerosol formation. Environmental Science & Technology, 31(7), 1890-1897, 1997.

Park, S., Bretherton, C. S., and Rasch, P. J.: Integrating Cloud Processes in the Community Atmosphere Model, Version 5, J. Climate, 27, 6821-6856, 10.1175/Jcli-D-14-00087.1, 2014.

Randerson, J. T., Chen, Y., van der Werf, G. R., Rogers, B. M., and Morton, D. C.: Global burned area and biomass burning emissions from small fires, J. Geophys. Res.-Biogeo., 117, G04012, https://doi.org/10.1029/2012JG002128, 2012.

Riahi, K., Van Vuuren, D.P., Kriegler, E., Edmonds, J., O'neill, B.C., Fujimori, S., Bauer, N., Calvin, K., Dellink, R., Fricko, O. and Lutz, W.: The shared socioeconomic pathways and their energy, land use, and greenhouse gas emissions implications: an overview. Global Environmental Change, 42, pp.153-168, 2017.

Tosca, M. G., Randerson, J. T., and Zender, C. S.: Global impact of smoke aerosols from landscape fires on climate and the Hadley circulation, Atmos. Chem. Phys., 13, 5227-5241, 10.5194/acp-13-5227-2013, 2013.

Tost, H., Jöckel, P. and Lelieveld, J.: Lightning and convection parameterisations–uncertainties in global modelling. Atmospheric Chemistry and Physics, 7(17), 4553-4568, 2007.

Van der Werf, G.R., Randerson, J.T., Giglio, L., Collatz, G.J., Mu, M., Kasibhatla, P.S., Morton, D.C., DeFries, R.S., Jin, Y.V. and van Leeuwen, T.T.: Global fire emissions and the contribution of deforestation, savanna, forest, agricultural, and peat fires (1997–2009). Atmospheric chemistry and physics, 10(23), 11707-11735, 2010.

Zou, Y., Wang, Y., Ke, Z., Tian, H., Yang, J., and Liu, Y.: Development of a REgion-Specific ecosystem feedback Fire (RESFire) model in the Community Earth System Model, J. Adv. Model Earth Sy., https://doi.org/10.1029/2018MS001368, 2019.